# Shear-Enhanced Compaction Analysis of the Vaca Muerta Formation

José G. Hasbani [1,]*[ID], Evan M. C. Kias [2][ID], Roberto Suarez-Rivera [2] and Victor M. Calo [3][ID]

1    Vista Energy, Av. del Libertador 101 Piso 12, Vicente lópez, Buenos Aires 1638, Argentina
2    W.D. Von Gonten Engineering, 10496 Katy Freeway, Suite 200, Houston, TX 77043, USA;
     evan.kias@wdvgco.com (E.M.C.K.); roberto@wdvglab.com (R.S.-R.)
3    School of Electrical Engineering, Computing and Mathematical Sciences, Curtin University, Engineering
     Pavillion Stg2 Room 317, GPO Box U1987, Perth, WA 6845, Australia; victor.calo@curtin.edu.au
*    Correspondence: jose.hasbani@vistaenergy.com

**Abstract:** The laboratory measurements conducted on Vaca Muerta formation samples demonstrate stress-dependent elastic behavior and compaction under representative in situ conditions. The experimental results reveal that the analyzed samples display elastoplastic deformation and shear-enhanced compaction as primary plasticity mechanisms. These experimental findings contradict the expected linear elastic response anticipated before brittle failure, as reported in several studies on the geomechanical characterization of the Vaca Muerta formation. Therefore, we present a comprehensive laboratory analysis of Vaca Muerta formation samples showing their nonlinear elastic behavior and irrecoverable shear-enhanced compaction. Additionally, we calibrate an elastoplastic constitutive model based on these experimental observations. The resulting model accurately reproduces the observed phenomena, playing a pivotal role in geoengineering applications within the energy industry.

**Keywords:** rock mechanics; numerical plasticity; Vaca Muerta

## 1. Introduction

Reservoir rocks' nonlinear and heterogeneous nature is typically simplified when analyzing deformation and failure during oil and gas well operations, such as drilling, hydraulic fracturing, and production [1]. Consequently, geomechanical engineers routinely assume that the material response is linear elastic until reaching brittle failure; this assumption allows them to use straightforward analytical approximations to model stress distribution around a wellbore [2]. These oversimplifications are widely used to solve wellbore stability and hydraulic fracture [3] problems. Typically, the linear elasticity assumption in unconventional reservoirs has been justified based on their brittleness [4], implying that reservoir rocks favorable to hydraulic fracturing treatments could be accurately modeled using linear elastic fracture mechanics [5]. This practice generally produces acceptable results when the reservoir rock exhibits a linear elastic response during laboratory experiments. However, when the mechanical behavior of these rocks departs from linear elasticity, more sophisticated theories need to be considered.

The Vaca Muerta Formation is the most prominent unconventional hydrocarbon reservoir located in the Neuquén Basin in Argentina. This shale rock is composed of marine sedimentary material rich in organic mudstones, limestones, and marls, and it was deposited in a distal ramp [6] during the Jurassic period. The mineralogy of this type of reservoir rock is dominated by calcite, quartz, mica, pyrite, and clays. Typically, the Vaca Muerta mudrock is described as a linear elastic material [7] in rate-independent mechanical problems or as a visco-elastic material [8] in geological time-dependent basin modeling problems. This reservoir can be found at approximately 2900 m depth, depending on its location in the Basin. The depth and unusually high pore of the Vaca Muerta formation

make drilling and coring operations a significant challenge [9], turning the acquisition of rock samples into a valuable asset in their own right to any energy operator company.

After embarking on the demanding venture of drilling and coring 250 m from the Vaca Muerta formation, we conducted routine core characterization, which involved a limited series of drained triaxial tests to determine the elastic properties of the formation. These material properties are inputs in our reservoir geomechanics characterization modeling. Surprisingly, laboratory test outcomes on these samples consistently indicate a highly nonlinear material response associated with an unusual compaction enhancement under drained shear stress, a phenomenon not commonly described for the Vaca Muerta mudrock (see recent publications, e.g., [7,8]). This finding may explain specific observations during the completion and production of a wellbore, such as inefficient fracture initiation during hydraulic fracture operations, unexpected well productivity due to under-stimulated rock volume, or poor efficiency in proppant placement [10,11]. Additionally, [12] expands on the importance of incorporating the rocks' plastic response in estimating pressure limits to ensure compelling drilling operations.

The compaction of porous rocks is often explained as the closure of porosity due to increasing effective stress, assuming that the solid constituents have negligible compressibility compared to their pore system. Typically, this mechanism is a factor in the life cycle of conventional reservoirs in the form of permeability reduction [13] and subsidence [14]. In addition, the regional in situ stress state plays a crucial role in the rock's failure mode, as demonstrated in numerous studies found in the literature addressing the brittle–ductile transition [15–19]. At laboratory scales, as the confining pressure increases, rock failure evolves from void volume creation through the formation and opening of micro-cracks that finally coalesce into a macroscopic material discontinuity known as brittle faulting [20]. This strong localization pattern is the consequence of the accumulation of irreversible plastic volumetric strain from grain rearrangement, the collapse of the pore volume, or grain fragmentation, which is known in the geomechanics community as shear-enhanced compaction [21]. Plastic deformation in mudrocks is typically attributed to the abundance of organic matter and clay inside the reservoir rock matrix [22].

Additionally, high contents of these constituents are often related to long-term viscoplastic mechanical responses, and its numerical modeling is comprehensively treated in [23] and well understood. On the contrary, laboratory experiments conducted in this work on the Vaca Muerta mudrock samples suggest that the irrecoverable volumetric strain is mainly controlled by shear-enhanced compaction, possibly dominated by microscopic rigid displacements of the grains and resulting in a macroscopic porosity reduction, being a predominant mechanism of unrecoverable volumetric plastic deformation in the Vaca Muerta shale. Our experimental observations underscore the need for a cogent constitutive model that captures this intricate mechanical response and ultimately improves the reliability of geomechanical engineers' estimations on applications involving this defying shale rock.

In this work, we present a comprehensive analysis of our scarce but compelling rock samples; we also adopt a phenomenological macroscopic elastoplastic constitutive model that adequately captures the observed material response during laboratory routine tests and calibrates its constitutive parameters so that the geoengineering community can reliably use this model in their applications. From all available phenomenological constitutive models, we choose the modified Cam-Clay (MCC) model [24,25] as it adequately describes the main experimental observations while maintaining a simple mathematical structure widely implemented in numerous commercial simulators. Although the MCC model was initially developed to characterize soils' critical state, it was extended to other types of cohesive-frictional materials [26,27], showing its ample applicability range. Inspired by the work published in [28,29], we adopt an implicit integration algorithm and simulate a drained triaxial test response at the Gauss point level, comparing the numerical simulation with experimental data.

Finally, we structure this manuscript as follows: a brief introduction is covered in this section; Section 2 establishes the laboratory procedures that we use and shows experimental evidence of compaction enhanced by shear in Vaca Muerta; Section 3 addresses the methodology for calibrating the constitutive material parameters for the MCC model using a standard rock mechanics laboratory test program to capture the Vaca Muerta mudrock's compaction response adequately; Section 4 discusses the numerical framework to integrate the elastoplastic constitutive model properly and presents simulated results; Section 5 opens a discussion about our findings highlighting limitations and advantages of the proposed methodology; and Section 6 concludes this work and details future research avenues.

## 2. Experimental Evidence of Shear-Enhanced Compaction in Vaca Muerta Mudrock

The Vaca Muerta formation has been typically described as a linear elastic material. In addition, several publications limit the mechanical characterization of Vaca Muerta to linear elasticity (see [7,8]). This assumption invites geomechanics practitioners to use simplified analytical models to solve oil field-related geomechanical problems, such as wellbore stability and fracture propagation. Drilling wells in Vaca Muerta is challenging; its depth and highly over-pressurized environment hinder the possibility of extracting rock samples to study and adequately characterize their mechanical and petrophysical properties. In addition, due to the complex logistics and operational risks, the removal of a core is carefully planned and infrequent because of the reasons mentioned above, and due to the destructive nature of rock mechanics tests, the number of samples delved for geomechanical characterization is scarce. As a part of a multidisciplinary project, we carefully plan to extract a 250 m continuous core from the Vaca Muerta formation covering the main productive sections to perform a complete set of laboratory studies focused on characterizing this prominent shale oil reservoir. Among all the involved disciplines, we perform routine geomechanics tests on a finite amount of samples from that core to preserve material for future geological characterization. We extract five cylindrical samples at two different productive sections in Vaca Muerta to measure elastic properties correctly to calibrate a geomechanical model to assess wellbore stability problems. However, after testing these samples, we observe that the linear elasticity assumption for this rock is far from being accurate.

We ran our rock mechanics program at the W. D. Von Gonten Engineering laboratory facilities in Houston, Texas. This program performed a drained hydrostatic compression test to investigate the rock's response under drained isotropic compression and four drained triaxial compression tests for vertically oriented samples to estimate the rock's elastic parameters. Following ASTM D4543 [30] and D7012 [31] testing standards, each sample was tested in a modern servo-controlled triaxial GCTS RTR-1000 loading system (see Figure 1).

Laboratory tests were performed under room temperature and drained conditions using a Teflon isolating jacket, cantilever radial displacement gauge, and four equally spaced axial displacement transducers (LVDT). Before conducting triaxial tests, three hydrostatic cycles followed by a uniaxial strain compression test were performed before the triaxial test (see Figure 2). The initial cycles of hydrostatic compression consolidate each sample and minimize the effect of coring-induced microcracks and other artifacts. The uniaxial-strain test seeks to evaluate the evolution of elastic properties as a function of the confining pressure.

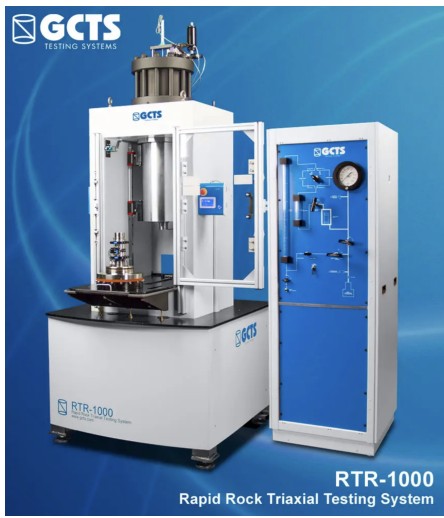

**Figure 1.** Triaxial GCST RTR-1000 testing system (https://www.gcts.com/).

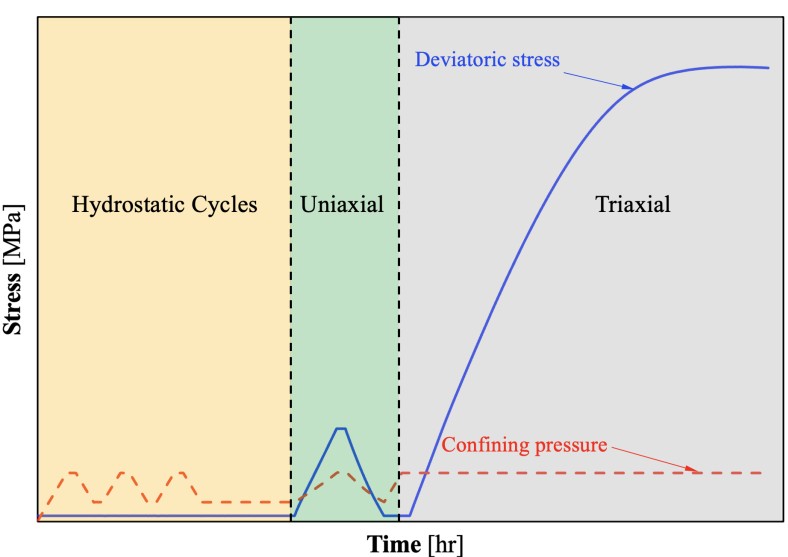

**Figure 2.** General laboratory test program for Vaca Muerta samples.

Table 1 shows the physical properties of each sample, where $D$ is the diameter, $L$ is the length, $V^0$ is the initial total volume, $m$ is the mass, $\phi^0$ is the initial porosity, $V_v^0$ is the initial pore volume, and $V_s^0$ is the initial solid volume.

**Table 1.** Vaca Muerta samples' physical properties.

| Properties\ID | Sample 0 | Sample 1 | Sample 2 | Sample 3 | Sample 4 |
|---|---|---|---|---|---|
| $D$ [cm] | 2.54 | 2.53 | 2.55 | 2.53 | 2.54 |
| $L$ [cm] | 5.05 | 5.09 | 5.12 | 5.05 | 5.06 |
| $V^0$ [cm$^3$] | 25.70 | 25.77 | 26.18 | 25.54 | 25.662 |
| $m$ [g] | 61.03 | 62.24 | 62.19 | 62.75 | 59.67 |
| $\phi^0$ [%] | 9.67 | 12.29 | 11.03 | 9.84 | 12.30 |
| $V_v^0$ [cm$^3$] | 2.48 | 3.16 | 2.88 | 2.50 | 3.14 |
| $V_s^0$ [cm$^3$] | 23.21 | 22.59 | 23.30 | 23.02 | 22.49 |
| Test type | Hydrostatic | Triaxial | Triaxial | Triaxial | Triaxial |

Several methods exist to determine a rock sample's initial porosity, $\phi^0$. For example, in [32], the authors proposed a novel technique relying on sample saturation and buoyancy. This methodology determined the effective porosity of dolomitic carbonates with karstic

voids. However, Vaca Muerta is a significantly low-permeability shale rock, and its porosity origins vary from embedded interconnected pores in an organic-rich phase to inter-granular connected pores [6], making the technique mentioned above for determining porosity impracticable. In [33], the authors proposed a novel methodology using nuclear magnetic resonance (NMR) spectroscopy and demonstrated its accuracy in determining total porosity in shale rocks. Given the similarity of the type of rock and the accuracy of this method, we used NMR spectroscopy to determine $\phi^0$. Following [34], we determined the initial solid and pore volume. We used the additive decomposition of the total initial volume $V^0$ in a solid $V_s^0$ and void $V_v^0$ contributions as in Figure 3, where

$$V^0 = V_s^0 + V_v^0. \tag{1}$$

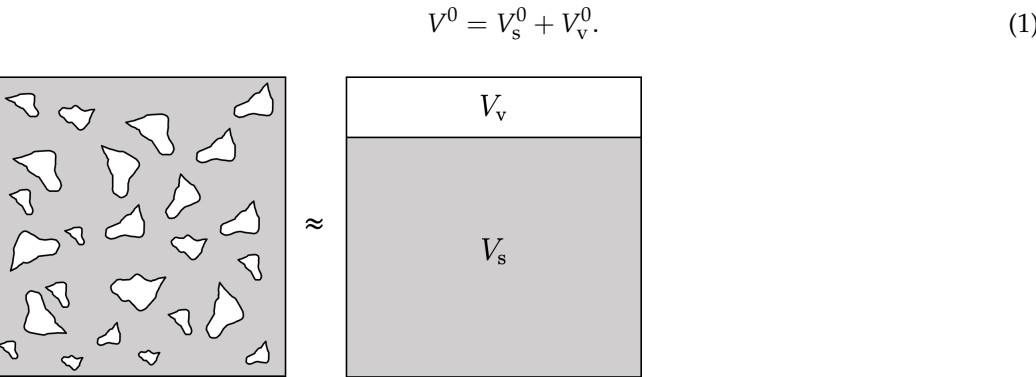

**Figure 3.** Additive decomposition of a porous medium volume.

In each test, we needed to estimate the evolution of the porosity as a function of increasing mean stress, $p$. First, we defined the porosity, solid volume, and pore volume at a time $t$ as

$$\phi^t := \frac{V_v^t}{V^t}, \tag{2}$$

$$V_s^t := (1 - \phi^t)\, V^t, \tag{3}$$

$$V_v^t := \phi^t\, V^t. \tag{4}$$

The pore system was unpressurized since all our laboratory tests were conducted under drained conditions. Therefore, we considered the solid volume (matrix) incompressible compared to the pore volume; namely, we considered $V_s^0 = V_s^t = V_s$. Under this assumption and applying additive decomposition to the initial and final volume, (3) and (4), the volumetric deformation at time $t$ adopts the following form:

$$\varepsilon_v^t = \frac{V^t - V^0}{V^0} = \phi^t \frac{V^t}{V^0} - \phi^0$$

where $V^0$, $\phi^0$, and $V^t$, $\phi^t$ are the initial volume and porosity and the porosity and volume at time $t$, respectively. During the test, we calculated $\varepsilon_v^t$, $\phi^0$, $V^0$ and $V^t$ from LVDT measurements. We, thus, can estimate the evolution of the porosity as follows:

$$\phi^t = \left(\varepsilon_v^t + \phi^0\right) \frac{V^0}{V^t}. \tag{5}$$

Before performing triaxial compression tests to estimate the elastic properties, we conducted a drained hydrostatic compression test to characterize the consolidation properties of the Vaca Muerta shale rock in Sample 0. Figures 4 and 5 show the evolution of the volumetric strain and the porosity as a function of the mean stress under drained isotropic compression during loading and unloading. Under loading conditions, we observe a monotonic linear evolution of the volumetric strain and a linear decreasing trend in porosity (see blue dashed line in Figures 4 and 5). This illustrates that the sample is consolidating under

hydrostatic loading. During unloading, we observe a residual volumetric strain and a reduction in porosity. This observation is meaningful since it corroborates the occurrence of compaction due to a pore volume reduction of approximately 0.5%, similar to the residual volumetric strain, reinforcing the assumption of matrix incompressibility under drained loading conditions.

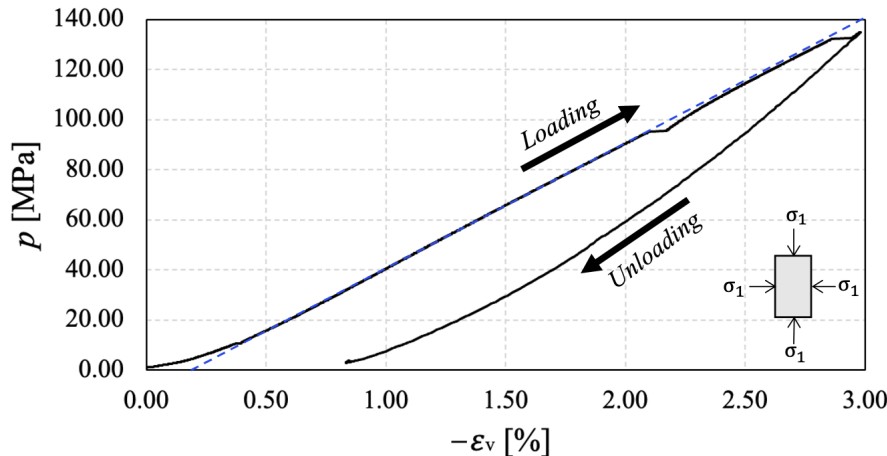

**Figure 4.** Evolution of volumetric strain as a function of mean stress during drained hydrostatic test on Sample 0.

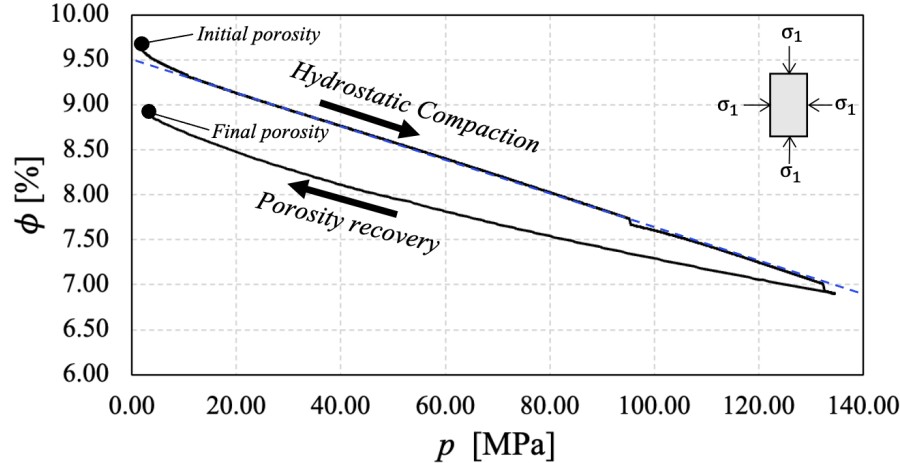

**Figure 5.** Evolution of porosity as a function of mean stress during drained hydrostatic test on Sample 0.

After the consolidation analysis of a representative Vaca Muerta sample, we conducted a series of drained triaxial compression tests on four samples to investigate their elastic properties and post-elastic stress–strain behavior. Several novel methodologies based on machine learning techniques are available in the literature to determine a rock's elastic constants. For instance, in [35], the authors proposed a comprehensive study of regression models to determine rock samples' elastic properties, which could be adequate when many samples are available. However, due to the challenging drilling environment of the Vaca Muerta formation, recovering a core involves high drilling costs and logistics planning. Therefore, due to the limited number of samples available, we followed the ASTM D7012-23 [33] standards to determine elastic properties in four available Vaca Muerta rock samples. Since we aimed to characterize the mechanical response of Vaca Muerta at in situ effective stress conditions, each sample was subjected to a confinement pressure ($\sigma_c$) of 18 MPa. We determined the elastic bulk modulus for each sample by analyzing the cyclic hydrostatic stage during our rock testing program (see Figure 2). During the hydrostatic

cycling stage, the deviatoric stress $q = \sigma_1 - \sigma_3$ remained constant, and the mean stress $p = \frac{1}{3}(\sigma_1 + 2\sigma_3)$ monotonically increased from 1 MPa to 18 MPa. From this test, we can determine the initial volumetric modulus during unloading $K^0$ considering the slope of the unloading curve from $p$ against the $\varepsilon_v$ chart. Elastic properties, namely Young's modulus ($E$) and Poisson's ratio ($\nu$), were determined at the drained triaxial stage by determining the slope of the linear trend in stress difference ($\sigma_1 - \sigma_3$) and the radial strain ($\varepsilon_r$) against the axial strain ($\varepsilon_1$) charts, respectively.

Figures 6–9 show the determination of the elastic constants from laboratory tests, and Table 2 summarizes its values for samples 1 to 4. During unloading/reloading hydrostatic cycles, an insignificant amount of hysteresis develops. We disregarded this behavior when determining the elastic bulk modulus and considered only the first unloading cycle. The maximum mean stress during hydrostatic cycling should be the confinement pressure under triaxial conditions. Otherwise, the triaxial stress path may induce unwanted isotropic compression. Therefore, the sample might show a different response at failure.

**Table 2.** Elastic properties from hydrostatic cycles and drained triaxial tests.

| Properties\ID | Sample 1 | Sample 2 | Sample 3 | Sample 4 | Mean Value |
|:---:|:---:|:---:|:---:|:---:|:---:|
| $E$ [GPa] | 16.05 | 15.54 | 20.96 | 16.34 | **17.22** |
| $K^0$ [GPa] | 13.65 | 12.22 | 14.09 | 11.19 | **12.78** |
| $\nu$ | 0.186 | 0.168 | 0.196 | 0.162 | **0.178** |

Our rock mechanics laboratory results show elastic property values that align closely with those published in [7]. However, during triaxial testing, the Vaca Muerta samples exhibit a highly non-linear response after a slight elastic deformation range. To investigate this non-linearity source, we also analyzed the evolution of the volumetric strain and the porosity (using (5)) against the mean stress $p$.

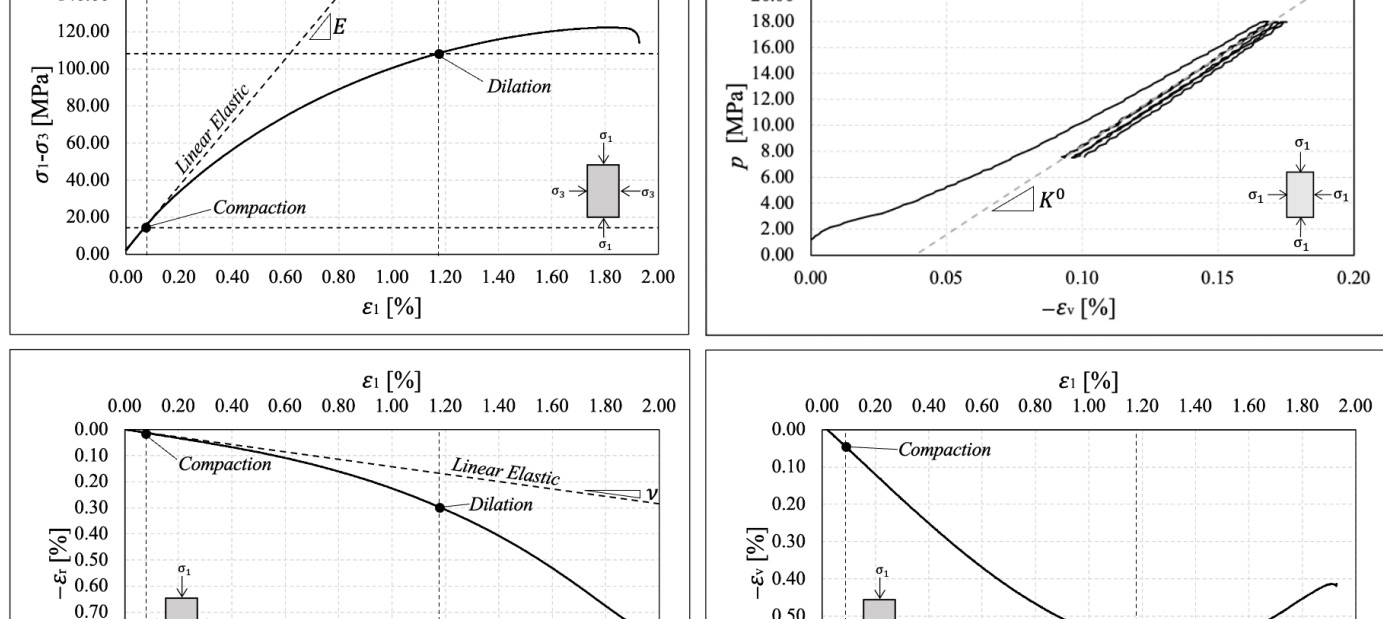

**Figure 6.** Elastic constant determination for Sample 1. (**Upper left**) Young's modulus determination. (**Upper right**) Bulk modulus determination. (**Bottom left**) Poisson's ratio determination. (**Bottom right**) Volumetric strain evolution during drained triaxial test.

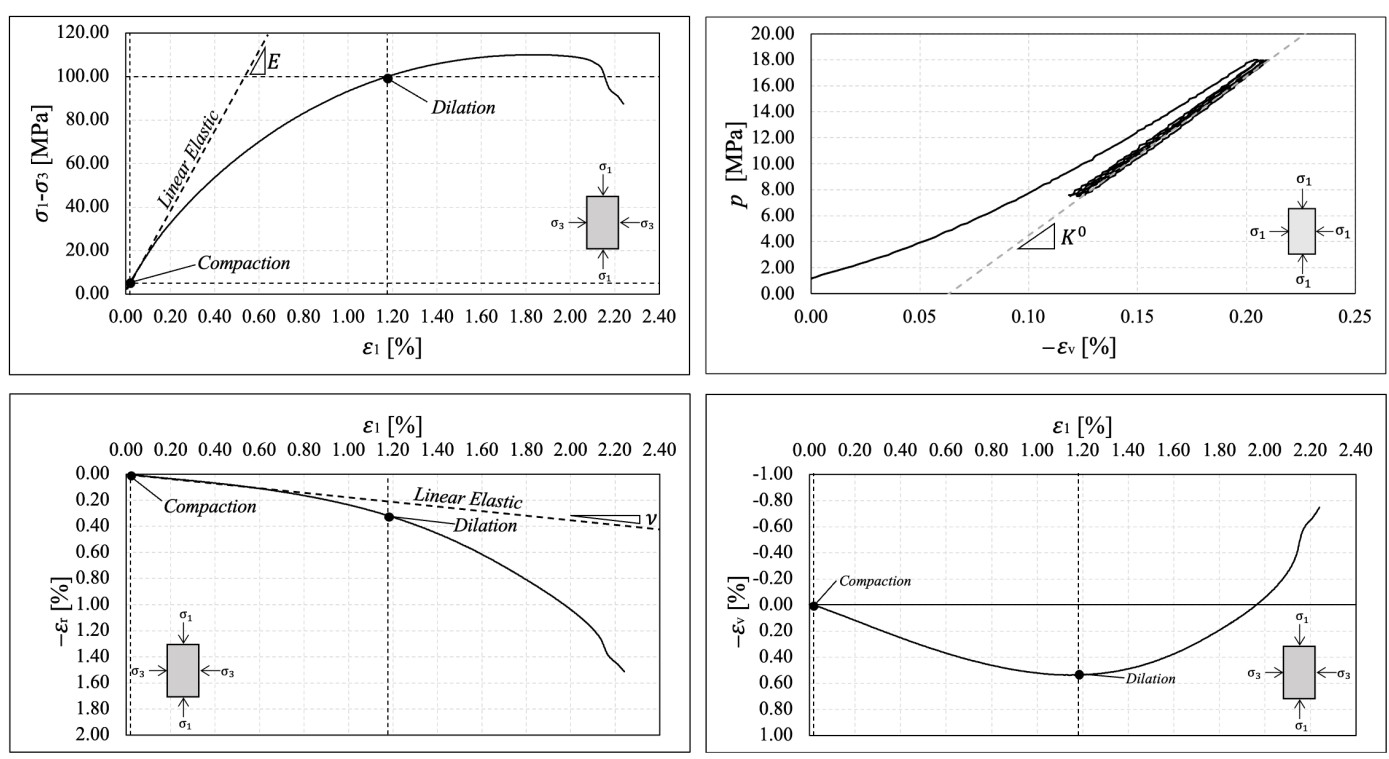

**Figure 7.** Elastic constant determination for Sample 2. (**Upper left**) Young's modulus determination. (**Upper right**) Bulk modulus determination. (**Bottom left**) Poisson's ratio determination. (**Bottom right**) Volumetric strain evolution during drained triaxial test.

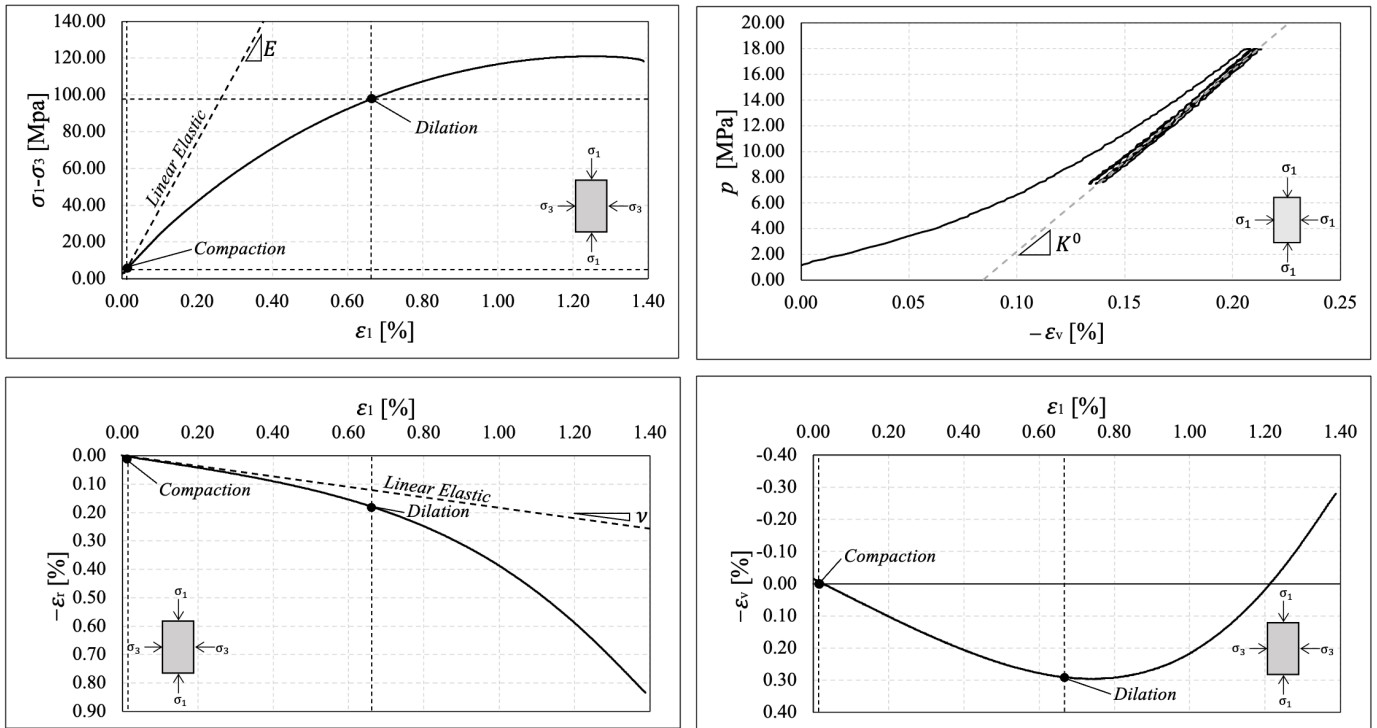

**Figure 8.** Elastic constant determination for Sample 3. (**Upper left**) Young's modulus determination. (**Upper right**) Bulk modulus determination. (**Bottom left**) Poisson's ratio determination. (**Bottom right**) Volumetric strain evolution during drained triaxial test.

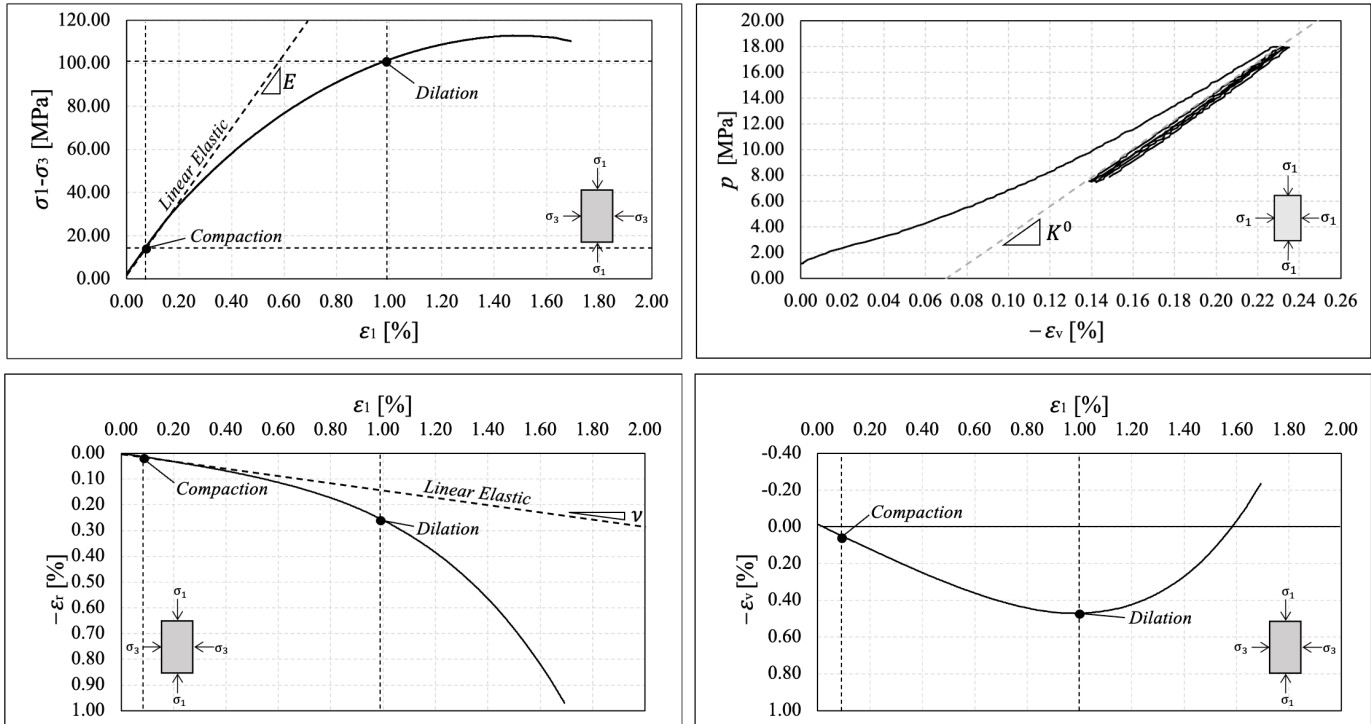

**Figure 9.** Elastic constant determination for Sample 4. (**Upper left**) Young's modulus determination. (**Upper right**) Bulk modulus determination. (**Bottom left**) Poisson's ratio determination. (**Bottom right**) Volumetric strain evolution during drained triaxial test.

Comparing the drained hydrostatic compression test response previously illustrated with the material's behavior under drained triaxial conditions, we observe significant deviation in the volumetric strain and porosity evolution from the observed hydrostatic compaction trend (see Figures 4 and 5). These experimental observations show that compaction is enhanced by shear until reaching dilatancy before localized failure. In Figures 10–13, we identify the mean stress value where the deviation from hydrostatic compaction trend occurs ($p_c^0$). This value is designated as the onset of compaction by [21]. According to [21], the deviation from the hydrostatic compaction trend line is a measure of the plastic contribution to the total volumetric strain, validating the decomposition of the total volumetric strain into an elastic and plastic component (see [34]), $\varepsilon_v = \varepsilon_v^e + \varepsilon_v^p$, under small strains. The same decomposition applies to the porosity reduction under small strains, $\Delta\phi$.

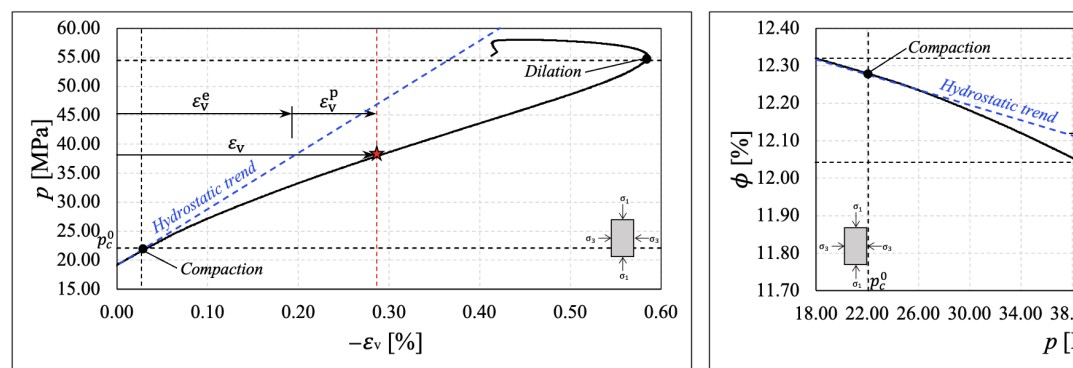
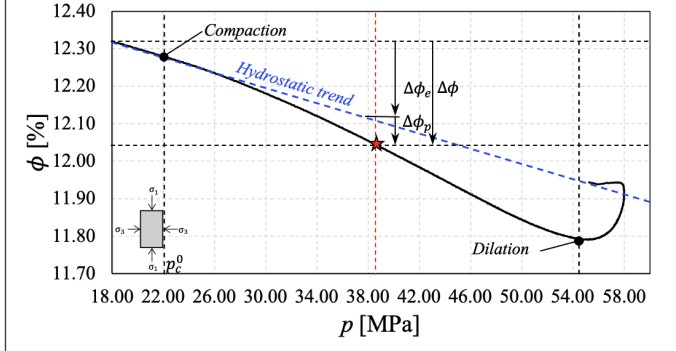

**Figure 10.** Shear-enhanced compaction analysis for Sample 1. (**Left**) Volumetric strain evolution as a function of mean pressure. (**Right**) Porosity evolution as a function of mean stress.

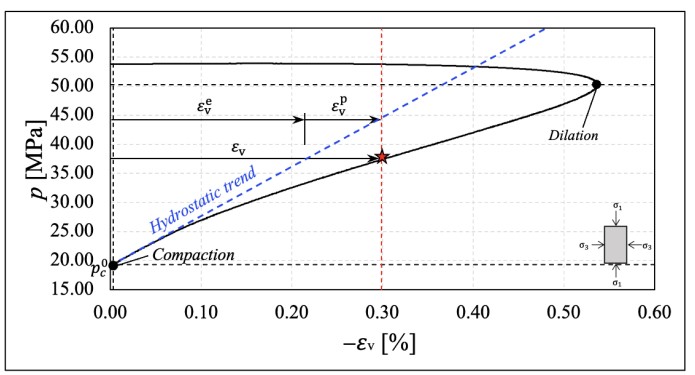
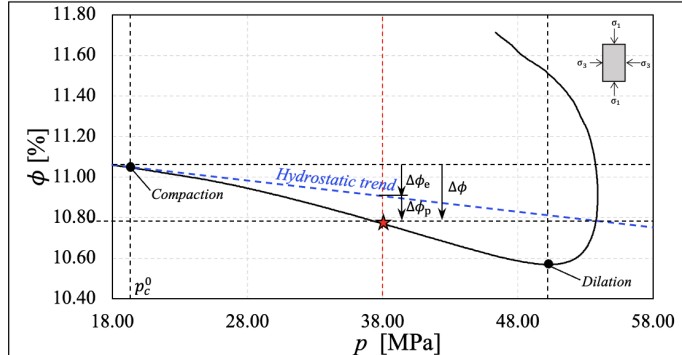

**Figure 11.** Shear-enhanced compaction analysis for Sample 2. (**Left**) Volumetric strain evolution as a function of mean pressure. (**Right**) Porosity evolution as a function of mean stress.

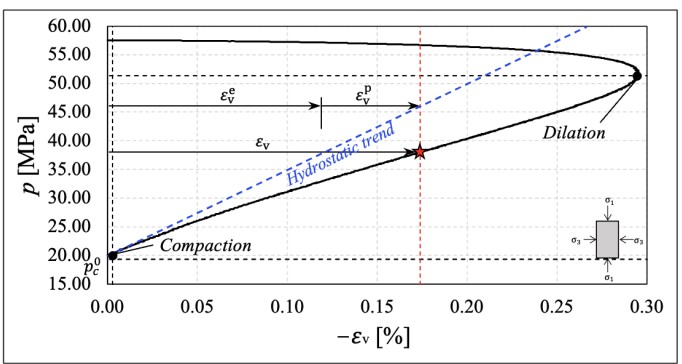
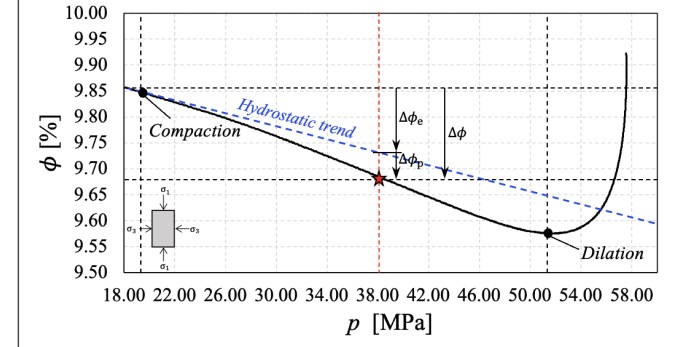

**Figure 12.** Shear-enhanced compaction analysis for Sample 3. (**Left**) Volumetric strain evolution as a function of mean pressure. (**Right**) Porosity evolution as a function of mean stress.

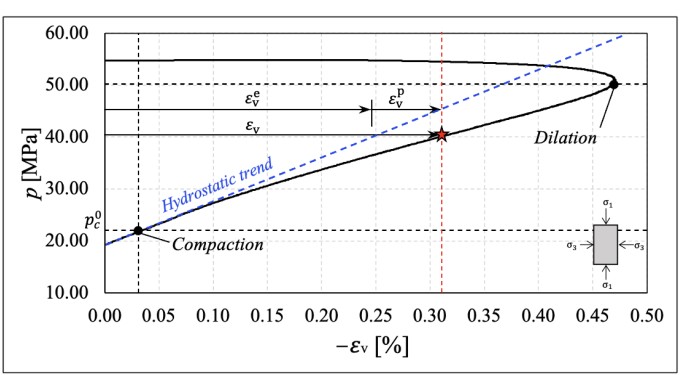
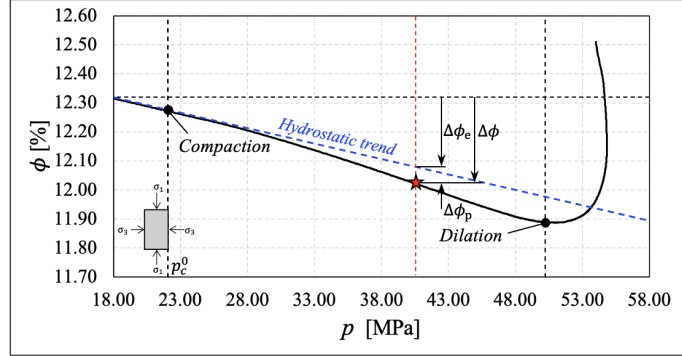

**Figure 13.** Shear-enhanced compaction analysis for Sample 4. (**Left**) Volumetric strain evolution as a function of mean pressure. (**Right**) Porosity evolution as a function of mean stress.

The experimental evidence of plastic deformation at confinement pressures comparable to the in situ conditions for Vaca Muerta samples is a remarkable finding that challenges the linear elasticity assumption in geomechanical engineering applications for drilling, fracturing, and producing wells targeting the Vaca Muerta formation. Moreover, our experimental results underscore the need to calibrate a more sophisticated constitutive model for geoengineering applications related to wellbore stability and fracture mechanics problems within the oil and gas sector.

## 3. Experimental Calibration of an Elastoplastic Constitutive Model for Vaca Muerta

In this section, we investigate adopting a macroscopic elastoplastic constitutive model capable of capturing shear-enhanced compaction. This mathematical framework should satisfy two main objectives. First and foremost, the constitutive parameters to calibrate the

chosen model need to be easily determined from standard laboratory tests. Secondly, to widely adopt this model among engineer practitioners, it should be comprehensive and commonly implemented in commercial finite element codes.

Various constitutive models were developed in the literature for modeling the elasto-plastic response of cohesive-frictional materials. Among the most popular yield criteria, we can highlight the Mohr–Coulomb (MC), Hoek–Brown (HB), and Drucker–Prager (DP) criteria in their inner and outer formulations (see [36]). Despite the wide application of the models above in failure analysis, these models have certain drawbacks in their numerical implementation. The MC yield criterion has discontinuities that need to be solved, the HB yield criterion possesses free parameters that are difficult to generalize, and the DP yield criteria do not capture various loading states typical for geomechanical applications (see [27,36]).

The modified Cam-Clay (MCC) yield criterion was initially developed to character-ize the elastoplastic response of wet clays. Recently, this yield criterion was extended to adequately capture the mechanical response of different materials ranging from pharma-ceutical applications [27] to the characterization of localization patterns in sandstones [26]. Additionally, in [23], a comprehensive analysis of the MCC yield criterion and its numerical implementation for capturing creep in Barnette shale is presented. This wide range of applications and its extensive adoption for modeling the elastoplastic response of cohesive-frictional materials drive us to explore its applicability to adequately capture the observed elastoplastic response in Vaca Muerta shale samples during drained triaxial testing.

### 3.1. Notation and Definitions

Throughout the rest of this manuscript, we extensively use tensor algebra. Therefore, we will briefly define our notation, meaningful identities, and tensor sign conventions.

We denote second-order tensors (i.e., matrix arrays that satisfy specific change of basis rules) using bold Greek symbols or letters; fourth-order tensors with upper case blackboard bold letters (e.g., $\mathbb{C}$)); vectors (i.e., first-order tensors) with lower case italic, bold letters; and scalars with lower case Greek symbols or letters. In addition, we adopt Einstein's summation convention (given $i = 1, 2, 3$; $a_i b_i = a_1 b_1 + a_2 b_2 + a_3 b_3$) to perform component-wise tensor operations, such as contractions (vector inner products) or double contractions (second-order tensor inner products). The symbol $\otimes$ denotes the tensor product (outer product).

We assume that all the operations are performed in Euclidean space. Thus, we make no distinction between covariant and contravariant basis vectors. Unit basis vectors in the Euclidean space $\mathbb{R}^3$ are denoted as $\mathbf{e}_i$, $i = 1, 2, 3$, with the properties $\mathbf{e}_i \cdot \mathbf{e}_j = \delta_{ij}$ and $\mathbf{e}_i \cdot \mathbf{e}_i = \delta_{ii} = 3$ for $i, j = 1, 2, 3$, where $\delta_{ij} \in \mathbb{R}^{3 \times 3}$ is the Kronecker Delta and where $\delta_{ii} = 1$ and $\delta_{ij} = 0$ if $i \neq j$. Therefore, we write first-order, second-order, and fourth-order tensors as follows:

$$\boldsymbol{a} = a_i \, \mathbf{e}_i, \quad \boldsymbol{\sigma} = \sigma_{ij} \, \mathbf{e}_i \otimes \mathbf{e}_j, \quad \mathbb{C} = C_{ijkl} \, \mathbf{e}_i \otimes \mathbf{e}_j \otimes \mathbf{e}_k \otimes \mathbf{e}_l, \quad \text{with} \quad i, j, k, l = 1, 2, 3$$

Since we work in a three-dimensional Euclidean space, the range for indexes will be omitted to avoid clutter in the rest of this manuscript.

The transpose of a second-order tensor $\boldsymbol{\sigma} \in \mathbb{R}^{3 \times 3}$ using index notation is defined as

$$\boldsymbol{\sigma}^T := \sigma_{ji} \, \mathbf{e}_i \otimes \mathbf{e}_j = \sigma_{ij} \, \mathbf{e}_j \otimes \mathbf{e}_i,$$

and the trace of a second-order tensor $\boldsymbol{\sigma}$ is given by $\text{tr}(\boldsymbol{\sigma}) := \boldsymbol{\sigma} : \mathbf{1}$, where $\mathbf{1} = \delta_{ij} \, \mathbf{e}_i \otimes \mathbf{e}_j$.

Given two vectors $\boldsymbol{a}, \boldsymbol{b} \in \mathbb{R}^3$ and two second-order tensors $\boldsymbol{\sigma}, \boldsymbol{\varepsilon} \in \mathbb{R}^{3 \times 3}$, the contrac-tion and the double contraction are given as follows:

$$\boldsymbol{a} \cdot \boldsymbol{b} := a_i b_j \quad \text{(Contraction)} \qquad \boldsymbol{\sigma} : \boldsymbol{\varepsilon} := \sigma_{ij} \, \varepsilon_{kl} \quad \text{(Double Contraction)}$$

We often make use of the symmetric fourth-order unit tensor, defined as

$$\mathbb{I} := \mathrm{I}_{ijkl}\, \mathbf{e}_i \otimes \mathbf{e}_j \otimes \mathbf{e}_k \otimes \mathbf{e}_l := \frac{1}{2}\left(\delta_{ik}\,\delta_{jl} + \delta_{il}\,\delta_{jk}\right)\mathbf{e}_i \otimes \mathbf{e}_j \otimes \mathbf{e}_k \otimes \mathbf{e}_l. \tag{6}$$

The symmetric fourth-order unit tensor admits the following decomposition:

$$\mathbb{I} := \mathbb{I}_{\mathrm{d}} + \mathbb{I}_{\mathrm{v}},$$

where $\mathbb{I}_{\mathrm{v}}$ and $\mathbb{I}_{\mathrm{d}}$ are the volumetric and deviatoric contributions given by

$$\mathbb{I}_{\mathrm{v}} := \frac{1}{3}\mathbf{1} \otimes \mathbf{1}, \qquad\qquad \mathbb{I}_{\mathrm{d}} := \mathbb{I} - \mathbb{I}_{\mathrm{v}}.$$

The operator $\| * \| : * \mapsto \mathbb{R}$ denotes the Frobenius norm for either vectors or second-order tensors. For vectors, $\|\boldsymbol{a}\| = \sqrt{\boldsymbol{a} \cdot \boldsymbol{a}} = \sqrt{a_i\, a_i}$, whereas for second-order tensors, $\|\boldsymbol{\sigma}\| = \sqrt{\boldsymbol{\sigma} : \boldsymbol{\sigma}} = \sqrt{\sigma_{ij}\, \sigma_{ij}}$. We usually work with variables that evolve during a pseudo-time increment $\Delta t = [t_{n+1},\, t_n]$, $\forall n \in \mathbb{N}^+$. The derivative of a tensor or scalar field concerning the pseudo-time $t$ is denoted as

$$\frac{\partial \boldsymbol{\sigma}(t)}{\partial t} := \dot{\boldsymbol{\sigma}}(t) = \dot{\sigma}_{ij}(t), \quad \forall \boldsymbol{\sigma}(t) \in \mathbb{R}^{3\times 3} \times [0,\, T] \qquad \frac{\partial \phi(t)}{\partial t} := \dot{\phi}(t), \quad \forall \phi : \mathbb{R}^n \times [0,\, T] \mapsto \mathbb{R}.$$

We adopt the rock mechanics convention in which compressive stresses are positive and tensile stresses are negative. In addition, we denote the effective Cauchy's stress tensor by $\boldsymbol{\sigma}$ and the infinitesimal strain tensor by $\boldsymbol{\varepsilon}$. The mean stress is $p := \frac{1}{3}\mathrm{tr}(\boldsymbol{\sigma}) = \boldsymbol{\sigma} : \mathbf{1}$, and the deviatoric effective stress $\mathbf{s} := \boldsymbol{\sigma} - p\,\mathbf{1}$ with the property that $\mathrm{tr}(\mathbf{s}) = 0$. Additionally, for a symmetric second-order tensor $\boldsymbol{\sigma}$, we define the following tensor invariants:

$$I_1(\boldsymbol{\sigma}) := \boldsymbol{\sigma} : \mathbf{1} = \sigma_{ii}, \tag{7}$$

$$I_2(\boldsymbol{\sigma}) := \frac{1}{2}\left(I(\boldsymbol{\sigma})_1^2 - \boldsymbol{\sigma} : \boldsymbol{\sigma}\right), \tag{8}$$

$$I_3(\boldsymbol{\sigma}) := \frac{1}{6}\left[2\,(\boldsymbol{\sigma} \cdot \boldsymbol{\sigma}) : \boldsymbol{\sigma} - 3\,I_1(\boldsymbol{\sigma})(\boldsymbol{\sigma} : \boldsymbol{\sigma}) + I_1^3\right) = \det(\boldsymbol{\sigma}). \tag{9}$$

The stress invariants for the deviatoric symmetric second-order tensor are

$$J_1(\mathbf{s}) := \mathrm{tr}(\mathbf{s}) = 0, \tag{10}$$

$$J_2(\mathbf{s}) := \frac{1}{2}\mathbf{s} : \mathbf{s}, \tag{11}$$

$$J_3(\mathbf{s}) := \frac{1}{3}(\mathbf{s} : \mathbf{s}) : \mathbf{s}. \tag{12}$$

Finally, the deviatoric stress invariants relate to the stress invariants as follows:

$$J_2(\mathbf{s}) = \frac{1}{3}\left(I_1^2(\boldsymbol{\sigma}) - 3\,I_2(\boldsymbol{\sigma})\right), \tag{13}$$

$$J_3(\mathbf{s}) = \frac{1}{27}\left(2\,I_1^3(\boldsymbol{\sigma}) - 9\,I_1(\boldsymbol{\sigma})\,I_2(\boldsymbol{\sigma}) + 27\,I_3(\boldsymbol{\sigma})\right), \tag{14}$$

$$I_2(\boldsymbol{\sigma}) = \frac{1}{3}\left(I_1^2(\boldsymbol{\sigma}) - 3\,J_2(\mathbf{s})\right), \tag{15}$$

$$I_3(\boldsymbol{\sigma}) = \frac{1}{27}\left(I_1^3(\boldsymbol{\sigma}) - 9\,I_1(\boldsymbol{\sigma})\,J_2(\mathbf{s}) + 27\,J_3(\mathbf{s})\right). \tag{16}$$

### 3.2. Continuous Elastoplastic Modified Cam-Clay Constitutive Model

Following [36], elastoplastic constitutive models should be able to capture the following experimental observations: for loads below a threshold that defines a flow criterion,

the material response is reversible (elastic); once the material reaches the limit condition, the deformation becomes partly irrecoverable (plastic); plastic deformations evolve the failure state as described by a hardening law; during unloading, the material response is elastic; the material response during the whole deformation process is quasi-stationary, and the material is thermodynamically stable (see Appendix A for a concise treatment of plasticity theory).

We describe the material's elastic response by adopting a linear Hookean model given by

$$\boldsymbol{\sigma} = \mathbb{C}^{\mathrm{e}} : \boldsymbol{\varepsilon}^{\mathrm{e}} = \mathbb{C}^{\mathrm{e}} : (\boldsymbol{\varepsilon} - \boldsymbol{\varepsilon}^{\mathrm{p}}), \tag{17}$$

where $\mathbb{C}^{\mathrm{e}}$ is the fourth-order elasticity tensor defined by the bulk modulus $K$, and the shear modulus $G$ is

$$\mathbb{C}^{e} := K\, \mathbf{1} \otimes \mathbf{1} + 2\, G \left( \mathbb{I} - \frac{1}{3} \mathbf{1} \otimes \mathbf{1} \right). \tag{18}$$

The elastic bulk and shear moduli, $K$ and $G$, are assumed to depend linearly on the mean stress $p$ (see [28]). The original expression of $K$ proposed in [28] depends on the void ratio $e$, which is defined as

$$e^{t} := \frac{V_{\mathrm{v}}^{t}}{V^{t}} \tag{19}$$

Combining (2) with (19), the void ratio related to the total porosity is as follows:

$$e^{t} = \frac{\phi^{t}}{1 - \phi^{t}} \tag{20}$$

Thus, we rewrite the expressions proposed in [28] in terms of the total porosity as

$$K := \frac{(1 + e^{t})\, p}{\kappa} = \frac{p}{\kappa(1 - \phi^{t})}, \tag{21}$$

$$G = \frac{3\, K(1 - 2\nu)}{2(1 + \nu)}, \tag{22}$$

where $\kappa$ is the volumetric deformation recovery, $\phi^{t}$ is the total porosity, and $\nu$ is Poisson's ratio.

The coupling of elastic shear and volumetric moduli may lead to energy dissipation under cyclic loading [28,29]. However, non-conservation is not an issue for monotonic loading, which is the main application in this manuscript. In addition, the definitions for $K$ and $G$ in (21) and (22) are widely used in practice for geoengineering applications (see [23,26,28,29]; thus, we adopt these expressions throughout the rest of this work. Additionally, we model the porosity evolution during loading using the following state equation in rate form:

$$\dot{\phi} = -\psi\, \dot{\varepsilon}_{\mathrm{v}}, \tag{23}$$

where $\psi$ represents the rock sample's porosity degradation rate during isotropic compression (hydrostatic compaction).

The mathematical description of an elastoplastic constitutive model involves the definitions of three main components [36,37] (see Appendix B for a detailed treatment). Firstly, the yield function describes the location of points where the material develops irrecoverable deformation. Secondly, the flow rule characterizes the evolution of plastic deformation. Finally, the hardening law typifies the evolution of the yield function throughout the plastic strain evolution.

Cam-Clay models [24,25] are widely used for plasticity characterization of the stress–strain response of cohesive-frictional materials subjected to three-dimensional stress states [26,27]. These simple models can realistically represent the compaction and dilation

responses of porous materials [23,28,29]. Cam-Clay models capture the typical pressure sensitivity and hardening of cohesive-frictional materials, requiring a few parameters that standard laboratory testing procedures can characterize (see [24,25]).

Recently, experimental data on limestone rocks that exhibit compaction and dilation during laboratory triaxial testing was reported in [38]. The same material response was observed in Vaca Muerta mudstone during drained triaxial testing reported in previous sections in this work. Thus, we selected a constitutive model that captures this nonlinear response for practical geomechanical applications. The MCC yield criterion is typically expressed in terms of the deviatoric stress $q$ and the mean stress $p$ (see Figure 14 for a schematic representation) as,

$$F_f(\boldsymbol{\sigma}) : \mathbb{R}^{3\times3} \mapsto \mathbb{R} \quad \text{such that} \quad F_f(\boldsymbol{\sigma}) = \frac{q^2}{M^2} + p(p - p_c), \tag{24}$$

where the deviatoric part of the effective stress tensor is defined as

$$q := \sqrt{\frac{3}{2}} \|\mathbf{s}\|, \tag{25}$$

and $M$ and $p_c$ define the critical state line (CSL) slope and the hardening parameter, respectively. The hardening parameter describes the yield surface's evolution as the mean stress increases during the material's deformation process.

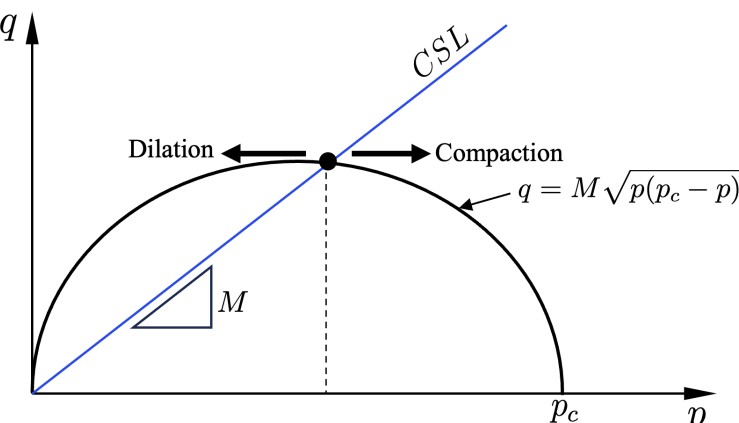

**Figure 14.** Representation of the Cam-Clay yield criterion in the $p - q$ plane.

Alternatively, we can express (24) in terms of the stress invariants as

$$F_f(\boldsymbol{\sigma}) = \frac{3}{2M} I_2(\boldsymbol{\sigma}) + \eta I_1^2(\boldsymbol{\sigma}) - \frac{1}{3} p_c I_1(\boldsymbol{\sigma}), \tag{26}$$

where

$$\eta = \frac{2M - 9}{18M}. $$

We adopt an associative flow rule (see Appendix B), which expresses the rate of plastic deformation as

$$\dot{\boldsymbol{\varepsilon}}^{\mathrm{P}} = \dot{\lambda} \frac{\partial F_f}{\partial \boldsymbol{\sigma}}, \tag{27}$$

where the flow direction is given by (see Appendix C.1)

$$\frac{\partial F_f(\boldsymbol{\sigma})}{\partial \boldsymbol{\sigma}} = \frac{1}{3}(2p - p_c)\mathbf{1} + \sqrt{\frac{3}{2}} \frac{2q}{M^2} \mathbf{n}, \tag{28}$$

in which $\mathbf{n} = \dfrac{\mathbf{s}}{\|\mathbf{s}\|}$ and $\dot{\lambda} > 0$ is the plastic multiplier.

**Remark 1** (Non-associativity). *Although associative flow rules characterize the plastic flow of granular materials, non-associative flow rules on the volumetric component of $\dot{\varepsilon}^p$ could be more suitable for some rocks. Nevertheless, assuming associative flow to capture the plastic flow in Vaca Muerta reproduces the experimental observation without adding additional heuristics.*

We adopt the hardening law in rate form (see [29]) as

$$\dot{p}_c = \chi\, p_c\, \dot{\varepsilon}_v^P, \tag{29}$$

where $\dot{\varepsilon}_v^P = \dot{\varepsilon}^P : \mathbf{1}$ is the volumetric component of the plastic strain rate tensor and $\chi$ is given by

$$\chi = \left[ (1 - \phi^t)(\gamma - \kappa) \right]^{-1}. \tag{30}$$

In (30), $\gamma$ and $\kappa$ are material parameters representing compaction and bulk volumetric recovery.

### 3.3. Material Parameter Determination for Vaca Muerta Elastoplastic Constitutive Model

To comprehensively define a macroscopic elastoplastic constitutive model for characterizing the shear-enhanced compaction response observed in Vaca Muerta mudrock samples, we need to estimate the material parameters $\kappa$, $\gamma$, and $\psi$ and the critical state line slope $M$. All these parameters can be determined consistently by analyzing the evolution of total porosity during the various test stages in laboratory testing. Compaction and bulk volumetric recovery parameters were measured in the hydrostatic cycling phase. The required number of cycles applied during testing induced a linear trend from loading and unloading cycles. At this point, the sample was compacted to a certain level, where artifacts from the core sample extraction are mitigated. Figure 15 depicts the evolution of total porosity estimated as in (2) against the mean stress during hydrostatic cycling.

Additionally, the $\kappa$ and $\gamma$ material parameters were determined by analyzing the slopes of $\phi - p$ plots during loading and unloading cycles. We also estimated the initial hardening parameter $p_c^0$ from the intersection of the loading and unloading curves, which was coincident with the onset of compaction value (refer to Figures 10–13). The initial hardening parameter is crucial for properly integrating the hardening law. One noteworthy observation is that during the unloading/reloading cycles, an insignificant amount of hysteresis developed. We disregarded this effect for parameter interpretation, and only the first unloading response was considered to define each material parameter and the initial hardening parameter $p_c^0$.

The porosity degradation parameter $\psi$, defining the porosity state Equation (23), was estimated by analyzing the slope of the porosity evolution against the volumetric strain during drained triaxial tests. Figure 16 shows the porosity evolution as a function of the volumetric strain $\varepsilon_v$, illustrating a clear linear relationship in the total porosity's degradation while the sample continues its compaction process.

The critical state line (CSL) slope $M$, which fully defines the MCC yield function, was estimated by analyzing the evolution of effective deviatoric stress $q$ against the effective mean stress $p$ during drained triaxial testing. This involved identifying the mean stress at which each sample dilated (refer to Figures 10–13). Figure 17 illustrates the evolution of the MCC yield criterion along the experimental stress path for the Vaca Muerta mudstone samples. The slope of the stress path was 3, the theoretical slope for triaxial conditions. Initially, the samples were loaded to an initial state $(p^0, q^0)$ within the initial yield locus defined by the major axis $p_c^0$, the initial hardening parameter captured by hydrostatic cycles (refer to Figure 15). Under these stress conditions, the sample response was elastic.

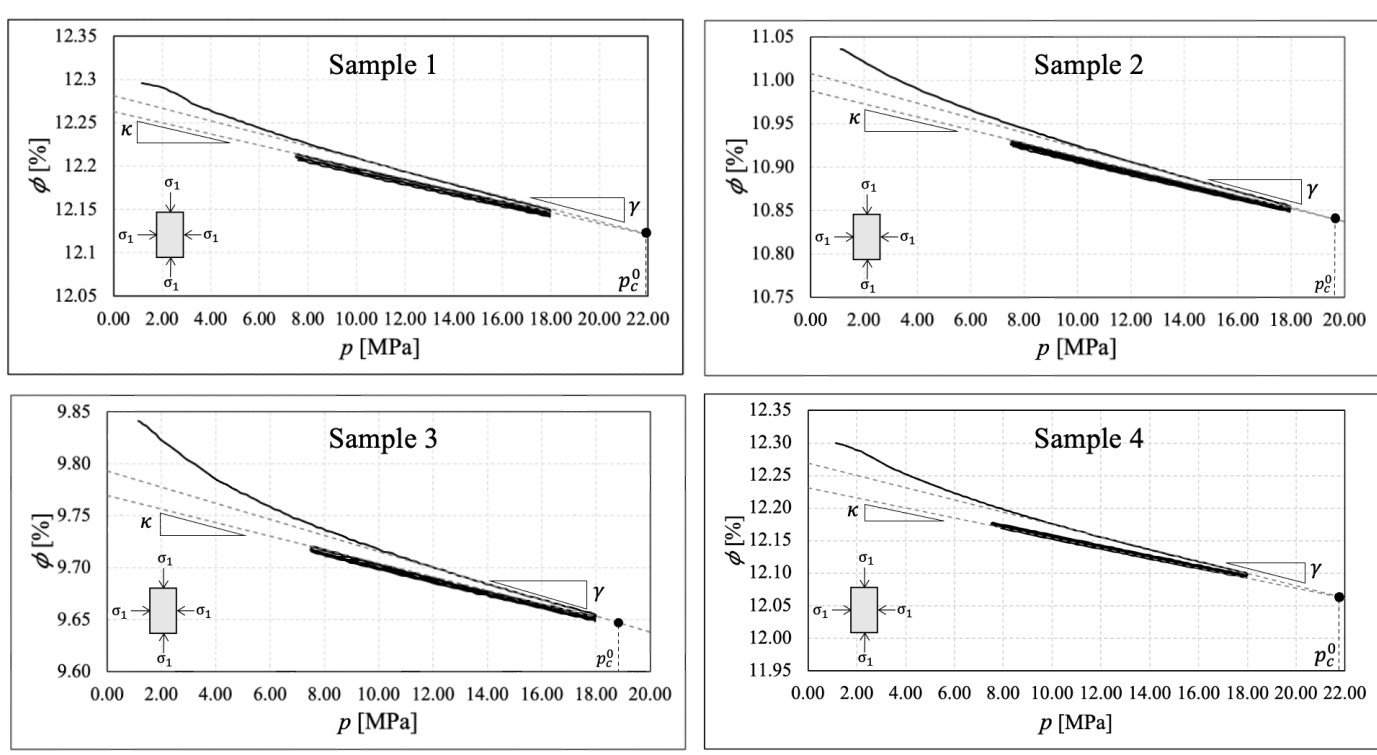

**Figure 15.** Compaction and bulk volumetric recovery parameters from hydrostatic cycling.

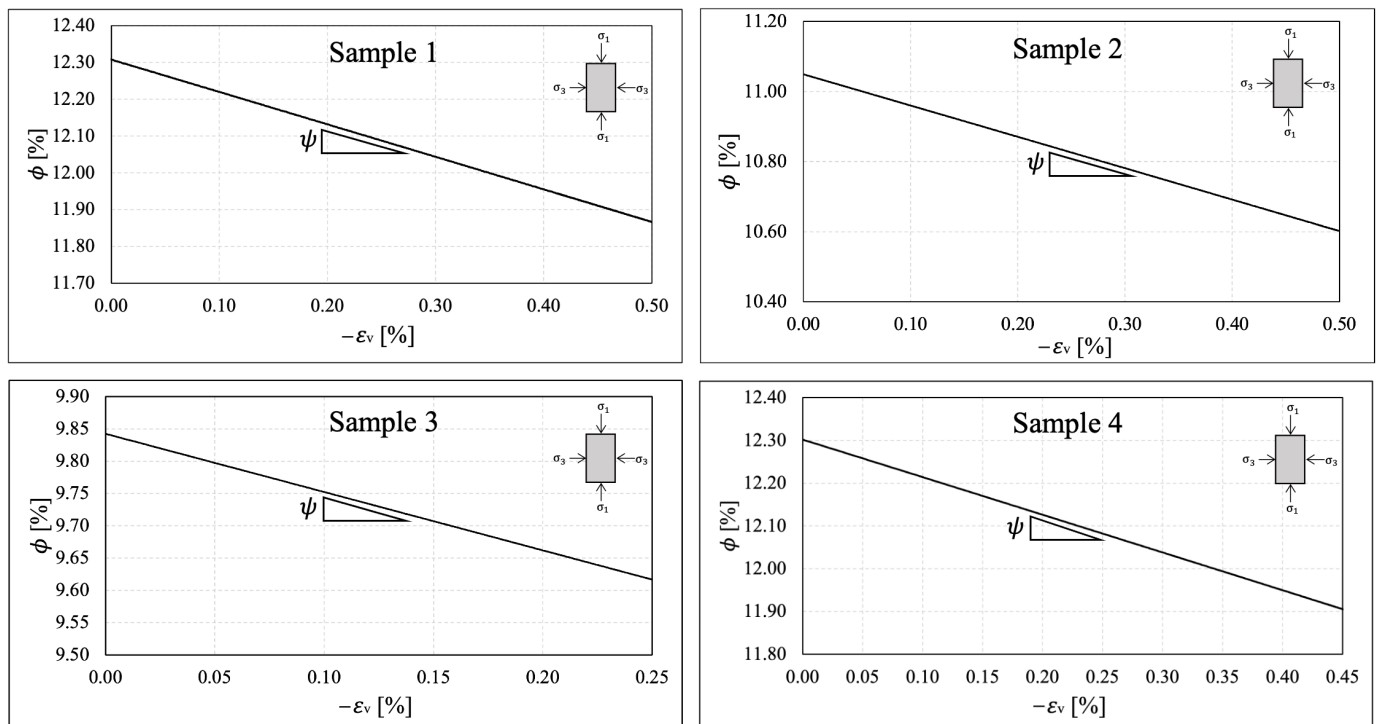

**Figure 16.** Porosity degradation parameter determination.

As the pair $(p, q)$ increased, the stress path reached its critical point, initiating dilation. At this juncture, the yield locus was defined by the critical hardening parameter $p_c^*$. With the onset of dilatancy, the samples progressively failed in shear, and the final Cam-Clay ellipse was defined by the final hardening parameter $p_c^t$. The CSL slope $(M)$ was then estimated by the intersection between the triaxial test stress path and the mean stress $p$ at which the sample increased in volumetric strain at constant pressure. This can be observed

in the $\varepsilon_v$ vs $p$, chart (see Figures 10–13). Thus, the CSL is a straight line from the origin to the latter intersection on the $q - p$ projection. We summarize the MCC material parameters for the Vaca Muerta mudstone based on the four samples we tested in Table 3.

**Table 3.** Modified Cam-Clay parameters for the Vaca Muerta formation.

| Properties\ID | Sample 1 | Sample 2 | Sample 3 | Sample 4 | **Mean Value** |
|---|---|---|---|---|---|
| $\kappa$ | 0.00123 | 0.00145 | 0.00156 | 0.00167 | **0.00147** |
| $\gamma$ | 0.00230 | 0.00270 | 0.00220 | 0.00250 | **0.00242** |
| $\psi$ | 0.0088 | 0.0089 | 0.0090 | 0.0088 | **0.0088** |
| $p_c^0$ [MPa] | 22.06 | 19.91 | 19.91 | 22.06 | **20.985** |
| $M$ | 2.02 | 1.98 | 1.98 | 2.00 | **1.995** |

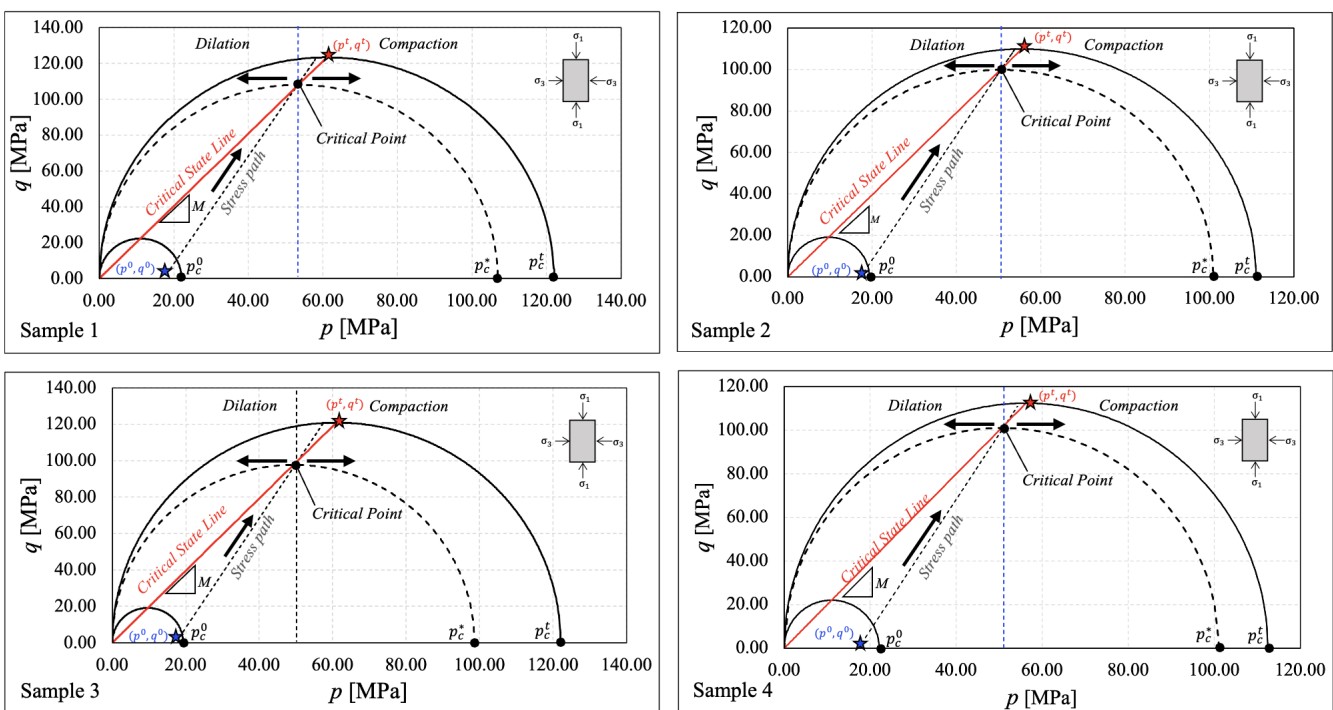

**Figure 17.** Determination of the critical state line slope $M$ from drained triaxial tests.

## 4. Numerical Integration of the Vaca Muerta Elastoplastic Constitutive Model

This section presents the numerical scheme for integrating the proposed elastoplastic constitutive model for its implementation in finite element codes. Although many geomechanically focused commercial codes include the MCC yield function to model the materials' non-linear response in engineering applications, we present a comprehensive treatment of the numerical formulation as we use it later in our numerical experiments. We based our implementation on the closest point projection mapping strategy, discussed in detail in [36,37] and adapted by [23,28,29] for updating the constitutive model state variables of the MCC model, relying on prediction-correction numerical methodology. We emphasize that the strategy presented here is a finite element framework building block, acting at the Gauss point level at a finite element.

We begin with the integration algorithm setting for the MCC elastoplastic constitutive model. Let $\Omega \in \mathbb{R}^d$, $d = 2, 3$. Consider the discretization $\Omega_h \subset \Omega$. Let us take an arbitrary Gauss point on a finite element $e_h \in \Omega_h$. Let $t \in [0, T]$, $T > 0$ be a pseudo-time and $n, k \in \mathbb{N}^+$ be the pseudo-time increment and iteration counters, respectively. The incremental strain tensor $\Delta\varepsilon_{n+1}^k$ is

$$\Delta\varepsilon_{n+1}^k := \varepsilon_{n+1}^k - \varepsilon_n. \tag{31}$$

Since the closest point projection mapping consists of a prediction-correction technique, we consider the trial state (elastic predictor) defined by freezing all the internal variables as

$$\hat{\boldsymbol{\sigma}}_{n+1} := \boldsymbol{\sigma}_n + \mathbb{C}^e : \Delta\boldsymbol{\varepsilon}^k_{n+1}, \tag{32}$$

$$\hat{p} := \frac{1}{3}\hat{\boldsymbol{\sigma}}_{n+1} : \mathbf{1}, \tag{33}$$

$$\hat{\mathbf{s}}_{n+1} := \hat{\boldsymbol{\sigma}}_{n+1} - \hat{p}_{n+1}\mathbf{1}, \tag{34}$$

$$\hat{q}_{n+1} := \sqrt{\frac{3}{2}}\|\hat{\mathbf{s}}_{n+1}\|. \tag{35}$$

where $\boldsymbol{\sigma}_n$ and $\boldsymbol{\varepsilon}_n$ are the converged effective stress and strain tensors of the previous pseudo-time step $n$. The return mapping (plastic correction) tensor equations in their general form are

$$\boldsymbol{\sigma}^k_{n+1} = \hat{\boldsymbol{\sigma}}_{n+1} - \mathbb{C}^e : \Delta\boldsymbol{\varepsilon}^P. \tag{36}$$

Integrating (A20) within $[t_n, t_{n+1}]$ leads to the following discrete plastic-strain increment, $\Delta\boldsymbol{\varepsilon}^P$:

$$\Delta\boldsymbol{\varepsilon}^P := \Delta\lambda\frac{\partial F_f}{\partial\boldsymbol{\sigma}}, \tag{37}$$

where $\Delta\lambda$ is the discrete consistency parameter. Consider the volumetric part of $\boldsymbol{\sigma}^k_{n+1}$:

$$p^k_{n+1} = \frac{1}{3}\boldsymbol{\sigma}^k_{n+1} : \mathbf{1} = \frac{1}{3}\hat{\boldsymbol{\sigma}}_{n+1} : \mathbf{1} - \frac{1}{3}\mathbb{C}^e : \Delta\boldsymbol{\varepsilon}^P : \mathbf{1} = \hat{p}_{n+1} - K_{n+1}\Delta\varepsilon^P_v, \tag{38}$$

where $\Delta\varepsilon^P_v = \Delta\boldsymbol{\varepsilon}^P : \mathbf{1} = \Delta\lambda\left(2p^k_{n+1} - p_c\right)$. Additionally, consider the deviatoric measure of $\boldsymbol{\sigma}^k_{n+1}$ (see Appendix C.2):

$$q^k_{n+1}(\Delta\lambda) = \frac{\hat{q}_{n+1}}{\left(1 + 6G_{n+1}\dfrac{\Delta\lambda}{M^2}\right)}. \tag{39}$$

Combining (38) and (39) results in the following system of scalar equations on $\Delta\lambda$:

$$p^k_{n+1}(\Delta\lambda) = \hat{p}_n + 1 - K\Delta\lambda\,(2\,p^k_{n+1} - p_c),$$
$$q^k_{n+1}(\Delta\lambda) = \frac{\hat{q}_{n+1}}{\left(1 + 6G\dfrac{\Delta\lambda}{M^2}\right)}. \tag{40}$$

The exact integration of the hardening law (29) gives:

$$p_c(\Delta\lambda) = (p_c)_n \exp\left(\chi\,\Delta\lambda\frac{\partial F_f}{\partial p^k_{n+1}}\right) = (p_c)_n \exp\left[\chi\,\Delta\lambda\left(2\,p^k_{n+1} - p_c\right)\right].$$

The consistency parameter $\Delta\lambda$ in (40)–(41) is calculated by imposing the consistency condition on $F_f(\Delta\lambda)$ (see Appendix B):

$$F_f(\Delta\lambda) = \left(\frac{q^k_{n+1}}{M}\right)^2 + p^k_{n+1}\left(p^k_{n+1} - p_c\right) = 0. \tag{41}$$

Since (41) couples the variables $p^k_{n+1}$ and $p_c$, $F_f(\Delta\lambda)$ cannot be evaluated explicitly. Therefore, $p^k_{n+1}$ and $p_c$ are updated iteratively by a local Newton–Raphson iteration. Thus, we rewrite the first equation in (40) as

$$p^k_{n+1} = \frac{\hat{p}_{n+1} + \Delta\lambda\,K_{n+1}\,p_c}{1 + 2\,\Delta\lambda\,K_{n+1}}, \tag{42}$$

and substituting (42) into the second equation in (40), we have the following scalar equation on $p_c$:

$$H_f(p_c) = (p_c)_n \exp\left(\chi \Delta\lambda \frac{2\,\hat{p}_{n+1} + \Delta\lambda\,K_{n+1}\,p_c}{1 + 2\,\Delta\lambda\,K_{n+1}}\right) - p_c = 0. \tag{43}$$

In Section 3.2, we define $K$ and $G$ as dependent on $p$ and $\phi^t$. Therefore, $K$ and $G$ are state variables updated at each strain increment during loading. We use an explicit integration scheme to update $\phi^t$, $K$, and $G$ as follows:

$$\phi_{n+1} = \phi_n - \psi(\Delta\varepsilon_v)_{n+1}, \tag{44}$$

$$K_{n+1} = \frac{p_{n+1}}{\kappa(1 - \phi_{n+1})}, \tag{45}$$

$$G_{n+1} = \frac{3K_{n+1}(1 - 2\nu)}{2(1 + \nu)}. \tag{46}$$

Thus, combining (31) with (46) and the derivatives of $F_f(\Delta\lambda)$ and $H_f(\Delta\lambda)$ with respect to the consistency parameter $\Delta\lambda$ (see Appendix C.3) to fully define the Newton–Raphson iteration scheme, we obtain the following closest point projection algorithm (see [28,37]) for updating $\sigma_{n+1}^k$, $\varepsilon_{n+1}^P$, and $p_c$ (see Figure 18 for a sketch of the closest point projection algorithm detailed in Algorithm 1):

---

**Algorithm 1** Closest point projection.

---

**1.** Given $\{\sigma_n,\ \varepsilon_n^P,\ \Delta\varepsilon,\ (p_c)_n\}$

**2.** Calculate the trial state:

$$\varepsilon_{n+1}^k = \varepsilon_n + \Delta\varepsilon_{n+1}^k,\quad \hat{\sigma}_{n+1} = \sigma_n + \mathbb{C}^e : \Delta\varepsilon_{n+1}^k,\quad \hat{p} = \frac{1}{3}\hat{\sigma}_{n+1} : \mathbf{1},$$

$$\hat{\mathbf{s}}_{n+1} = \hat{\sigma}_{n+1} - \hat{p}_{n+1},\quad \hat{q}_{n+1} = \sqrt{\frac{3}{2}}\|\hat{\mathbf{s}}_{n+1}\|.$$

**3.** Evaluate the modified Cam-Clay yield function in the trial state: $F_f[\hat{\sigma}_{n+1}, (p_c)_n]$

**4. IF** $F_f[\hat{\sigma}_{n+1}, (p_c)_n] < 0$:            ▷ Elastic Step

    $\sigma_{n+1}^k = \hat{\sigma}_{n+1},\quad \varepsilon_{n+1}^k = \varepsilon_n + \Delta\varepsilon_{n+1}^k,\quad \varepsilon_{n+1}^P = \varepsilon_n^P,\quad (p_c)_{n+1} = (p_c)_n.$

**5. ELSE IF** $F_f[\hat{\sigma}_{n+1}, (p_c)_n] \geq 0$:            ▷ Plastic Step

    i. Initialize:

        $\sigma_{n+1}^k = \sigma_n,\quad \varepsilon_{n+1}^p = \varepsilon_n^p,\quad (p_c)_{n+1} = (p_c)_n.$

    ii. **WHILE** $\left|F_f\left[\sigma_{n+1}^k, (p_c)_{n+1}\right]\right| > \text{FTOL}$:      ▷ Outer Newton–Raphson

      ○ **WHILE** $\left|H_f[(p_c)_{n+1}]\right| > \text{HTOL}$:      ▷ Inner Newton–Raphson

$$\star\ (p_c)_{n+1} = (p_c)_{n+1} - H_f[(p_c)_{n+1}]\left(\frac{\partial H_f}{\partial p_c}\right)^{-1}$$

      ○ $\Delta\lambda^k = \Delta\lambda^{k-1} - F_f\left[\sigma_{n+1}^k, (p_c)_{n+1}\right]\left(\frac{\partial F_f}{\partial \Delta\lambda}\right)^{-1}$

      ○ $p_{n+1}^k = \dfrac{\hat{p}_{n+1} + \Delta\lambda^k K p_c}{1 + 2\,\Delta\lambda^k K}$

      ○ $q_{n+1}^k = \dfrac{\hat{q}_{n+1}}{\left(1 + 6G\frac{\Delta\lambda^k}{M^2}\right)}$

**6. RETURN**:

$$\Delta\varepsilon_{n+1}^P = \Delta\lambda^k \frac{\partial F_f}{\partial \sigma},\quad \sigma_{n+1}^k = \hat{\sigma}_{n+1} - \mathbb{C}^e : \Delta\varepsilon_{n+1}^P,\quad \varepsilon_{n+1}^P = \varepsilon_n^P + \Delta\varepsilon_{n+1}^P$$

---

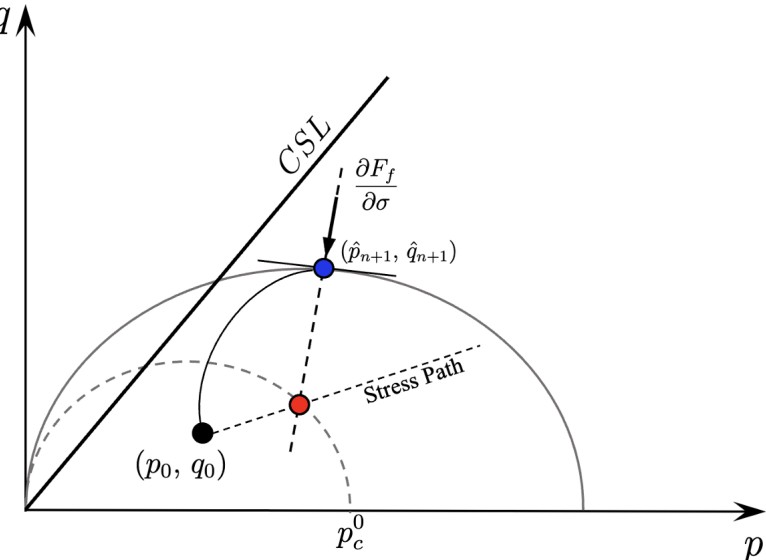

**Figure 18.** Internal variables updated by the closest point projection algorithm. The blue point is the trial state for the internal variables (elastic prediction). The red dot is the final state for the internal variables (plastic correction).

*Triaxial Test Simulation of Vaca Muerta Samples*

In this section, we evaluate the accuracy of the calibrated MCC model in reproducing compaction induced by shear in Vaca Muerta mudrock. Dilation is not included in our numerical framework, meaning the material response is perfectly plastic after reaching the critical compaction point. Our benchmark is focused on numerically reproducing a drained triaxial test using the calibrated model and comparing our simulated results against the presented experimental data.

The numerical experiment focuses on a single Gauss point $G_p$, inducing a triaxial stress state by considering an initial hydrostatic stress state at $G_p$ of the form

$$\boldsymbol{\sigma}_0 = \begin{bmatrix} \sigma_c & 0 & 0 \\ 0 & \sigma_c & 0 \\ 0 & 0 & \sigma_c \end{bmatrix} \mathbf{e}_i \otimes \mathbf{e}_j. \tag{47}$$

and applying the following deviatoric strain tensor:

$$\Delta \boldsymbol{\varepsilon}_0^d = \Delta \varepsilon \begin{bmatrix} 1 & 0 & 0 \\ 0 & 0 & 0 \\ 0 & 0 & 0 \end{bmatrix} \mathbf{e}_i \otimes \mathbf{e}_j - \frac{\Delta \varepsilon}{3} \begin{bmatrix} 1 & 0 & 0 \\ 0 & 1 & 0 \\ 0 & 0 & 1 \end{bmatrix} \mathbf{e}_i \otimes \mathbf{e}_j, \tag{48}$$

where $\sigma_{n+1}^k$ is the converged stress tensor in principal components (i.e., $\sigma_1$, $\sigma_2$, and $\sigma_3$, where $\sigma_1 \geq \sigma_2 \geq \sigma_3$). During the triaxial test, the stress components at $G_p$ must satisfy $\sigma_2 = \sigma_3 = \sigma_c$. Therefore, after each strain step, we iteratively enforced this condition by increasing the volumetric component of the deviatoric strain tensor increment $\Delta \boldsymbol{\varepsilon}_0^d$. In addition, we utilized a confinement pressure $\sigma_c$ of 18 MPa. This numerical experiment considered a strain-driven evolution considering a strain increment $\Delta \varepsilon$ of $8 \times 10^{-5}$.

We considered material parameters within the range we obtained for the Vaca Muerta mudstone samples. Therefore, we chose the average values for Poisson's ratio ($\nu$), compaction recovery ($\gamma$), bulk volume recovery ($\kappa$), porosity degradation ($\psi$), CSL slope ($M$), and initial onset compaction parameter ($p_c^0$), as reported in Table 2 and Table 3, respectively. We chose an initial porosity ($\phi^0$) of 12.3% and numerical tolerances for the return mapping convergence *FTOL* and hardening update *GTOL* of $1 \times 10^{-6}$.

In Figure 19, our triaxial test simulation utilizing the calibrated MCC yield function is compared with the drained triaxial laboratory test conducted on Vaca Muerta mudstone

samples. The calibrated constitutive model accurately reproduces the key features observed in our experimental laboratory tests. Additionally, this model effectively captures volumetric strain and unrecoverable deformation due to compaction induced by shear despite employing an associative flow rule for updating state variables.

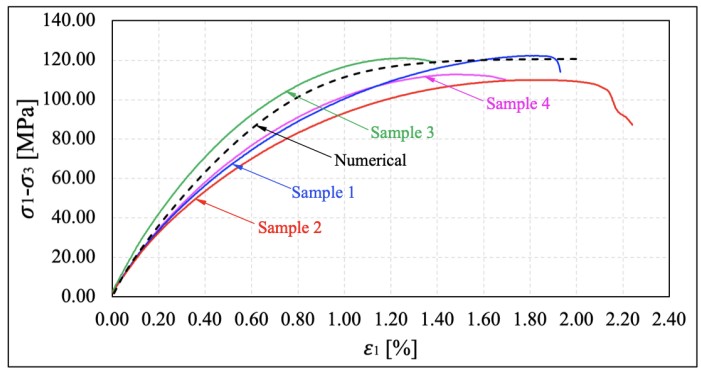 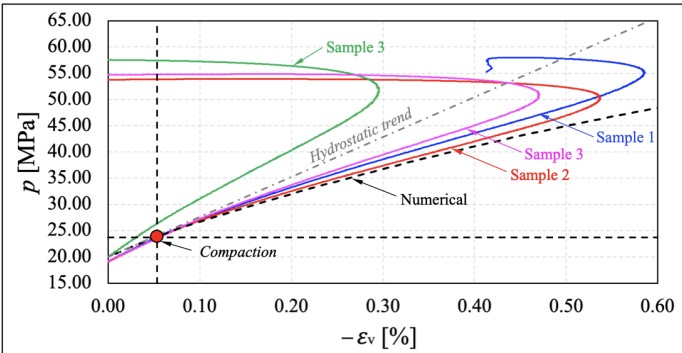

**Figure 19.** Comparison between numerical simulation of a triaxial test and laboratory tests. (**Left**) Deviatoric stress against axial strain. (**Right**) Mean stress against volumetric strain.

One limitation of this model is its inability to represent dilation response. The introduced state equations fail to capture this phenomenon, as the hardening law induces compaction as the yield function evolves. Furthermore, the model predicts perfect plasticity instead of dilation after the sample's critical deformation. Although some samples exhibited dilation and softening, these phenomena occurred at the end of the triaxial test, just before the shear failure of the sample. Thus, our model accurately captures the primary plastic response attributed to compaction.

## 5. Discussion

The rock mechanics characterization of Vaca Muerta has predominantly focused on its linear elastic response [7]. This assumption, applied in various engineering applications crucial for harnessing energy from this unconventional reservoir, simplifies the analysis of the stability at the wellbore wall during drilling [9] and the propagation of fractures during wellbore completion [3] for hydrocarbon production. However, conclusions drawn from such studies might be misleading, as evidenced by several field observations, including inefficient fracture initiation, proppant misplacement during hydraulic fracture operations, and unexpected well productivity due to under-stimulated rock volume [10,11]. These observations have often been attributed to uncertainties in determining the in situ stress state, typically assessed using a linear elastic constitutive model, either isotropic or anisotropic. This claim is frequently based on a lack of evidence of a mechanical response other than linear elasticity. Therefore, the experimental results presented in this manuscript are fundamental for advancing the characterization and modeling of Vaca Muerta's mechanical response.

This work is a starting point to motivate geomechanical practitioners to focus on the mechanical study of the Vaca Muerta formation to transition to more comprehensive rock mechanics models. This manuscript presents a thorough rock mechanics characterization of Vaca Muerta rock samples. One limitation of our laboratory study is the restricted number of samples, a common issue for various subsurface study groups in operator companies involved in the multidisciplinary characterization of this reservoir. This limitation prevents us from using more sophisticated techniques relying on machine learning regression methods (see [32] for a comprehensive treatment of this issue) to determine and find a generalized elastic constitutive model for Vaca Muerta. Additionally, our characterization of irrecoverable volumetric deformation relies on the evolution of total porosity as a function of volumetric strain. Therefore, the accurate determination of the initial porosity of each sample is crucial for estimating compaction and dilation evolution throughout

laboratory tests. Determining shale rock porosity is challenging due to the significantly low permeability of such rocks. Although recent research proposes indirect determination of porosity using saturation and buoyancy procedures (see [32] for further details), these techniques are based on the high permeability of the rock specimens used in the study, enabling full saturation. Unfortunately, saturating Vaca Muerta samples without altering their original pore structure is delicate. Given this limitation, we determine the initial porosity of each sample using NMR technology proposed in [33]. Over the last three years, this technique has been widely adopted to measure porosity in representative U.S. and Argentina shale rocks, becoming a standard methodology that petrophysical practitioners heavily rely on.

Beyond laboratory observations, we present a methodology to quantify shear-enhanced compaction properly, relying on a standard rock mechanics laboratory test program typically conducted by geomechanical practitioners and field experts. However, one possible limitation of this methodology is the macroscopic characterization of shale rock compaction relying on porosity degradation. Vaca Muerta is a fine-grained mudstone, and its solid constituents can be assumed to be incompressible compared to the pore volume when the deformation process is conducted under drained conditions at a macroscopic scale. However, this work does not cover undrained conditions or mesoscale and microscale modeling, which might be subjects of future studies. The degree of saturation in the pore system can also play an essential role in the deformation processes of cohesion frictional materials, as comprehensively treated in [34]; however, due to the drained conditions of our laboratory tests, this effect is not considered. Despite the limitations above, this work aims to spark discussion and expand efforts in the characterization of this unconventional reservoir rock in the Vaca Muerta geomechanics scientific community. Therefore, the observation of enhanced compaction during drained triaxial testing is a novelty that was not previously documented and not expected in Vaca Muerta.

After characterizing compaction enhancement driven by the shear stress path in our laboratory measurements, we endeavor to propose a constitutive model capable of reproducing this experimental observation. The choice of the MCC constitutive model is inspired by the compactive response of other cohesive-frictional materials like soils (see [24,25]). Although the MCC yield function was initially proposed for modeling wet clays, its application was extended beyond soil mechanics to various cohesive-frictional materials ranging from pharmaceutical powders to rocks (see [23,26,27]). In addition, the extension of the MCC constitutive model to predict viscoplastic response in a U.S. shale rock under long-term compressive loads is presented in [23]. The broad applicability of this constitutive model and its widespread implementation in various finite element commercial codes prompt us to provide a calibration procedure that involves estimating MCC material parameters from standard rock mechanics laboratory test programs.

Moreover, MCC yield function parameters relate directly to our laboratory observations, potentially enhancing its acceptance among practitioners. We focus on calibrating the MCC model to adequately capture compaction since it is the most significant response observed during our rock mechanics experiments. However, another possible limitation of this constitutive model is that dilation is not included in our mathematical framework, restricting its applicability solely to modeling compaction. Despite this limitation, the presented model can be widely applied to various geoengineering problems inside the compaction strain range.

The numerical framework presented in this work plays a crucial role in evaluating the performance of the MCC model to capture experimental observations accurately. The implementation adopted in this manuscript draws inspiration from pioneering works in [28,29]. An essential characteristic of our numerical integration strategy is using the closest point projection mapping algorithm, a standard procedure widely employed in numerical plasticity, enhancing its acceptance among geomechanics practitioners. Moreover, throughout the numerical experiments, we use an associated flow rule that perfectly aligns with the thermodynamics of deformable solids theory. This feature makes the model straightfor-

ward to implement and ensures thermodynamic consistency. Of particular significance is the observation from the numerical modeling perspective that we used an associated flow rule without over-predicting volumetric strains [36]. Typically, non-associated flow rules are adopted to capture compaction accurately. This often involves a combination of associativity in the deviatoric component and non-associativity in the volumetric component of the return mapping direction [36]. The latter approach might not be thermodynamically consistent and usually requires a dedicated verification of the Clasius–Duhem inequality (see Appendix A), leading to restrictive conditions on material parameters.

## 6. Conclusions and Future Work

This work presents experimental evidence of enhanced compaction response due to shear in Vaca Muerta mudstone, a phenomenon often overlooked and not characterized in standard rock mechanics laboratory procedures. The methodologies proposed in this manuscript aim to establish a consistent and self-contained procedure to accurately calibrate the MCC constitutive model, which is widely used in various geoengineering applications and recently extended for different cohesive-frictional materials beyond soils.

We calibrate a constitutive model capable of capturing the volumetric plasticity of Vaca Muerta mudstone using a standard laboratory testing program. The integration algorithm, inspired by [28], effectively updates the state variables $\sigma^k_{n+1}$ and $\varepsilon^p_{n+1}$. This algorithm employs a closest point projection strategy primarily proposed by [37], incorporating an associative flow rule and a simple hardening law. More importantly, while adopting an associative flow rule, the model accurately captures compaction without over-predicting volumetric strains. Our mathematical model does not aim to describe the dilatant behavior of this rock as the plastic load evolves and the algorithm updates the state variables. Therefore, the material behaves as perfectly plastic after a critical compaction value. The model effectively captures the main plastic dissipation mechanism due to compaction for Vaca Muerta mudstone samples, even though dilatancy was not included in the constitutive model formulation.

The calibrated MCC model and the integration algorithm we describe can be implemented in standard finite element routines to accurately capture the compaction of Vaca Muerta mudstone in more complex stress states and geomechanics applications, such as wellbore stability problems and hydraulic fracture propagation problems. This implementation has the potential to shed light on unexpected field observations related to hydrocarbon production performance, refined allowable pressure limits during drilling, and ineffective fracture propagation and proppant placement during hydraulic fracture operations. Additionally, geomechanics experts can utilize the material parameters of the calibrated MCC model in commercial finite element codes, allowing them to explore the impact of this material response in their engineering applications within the energy sector.

While the observations in this work provide a foundational understanding of the mechanical response of Vaca Muerta mudstone, future efforts will focus on incorporating dilatancy into the hardening law to capture this phenomenon accurately. Potential research avenues in this direction include implementing Rowe's dilatancy theory [39], widely used to capture softening in various cohesive-frictional materials. Additionally, after framing the mathematical framework to incorporate dilatancy, a complementary research path could extend the applicability of this model by studying localization at a constitutive level [26]. This could provide localization directions to model shear bands in Vaca Muerta shale rock properly. Developing a compelling localization theory would enhance the understanding of the necessary conditions for the occurrence of wellbore collapse, allowing drilling engineers to improve their drilling fluids, significantly impacting drilling performance.

**Author Contributions:** Conceptualization, J.G.H.; formal analysis, J.G.H.; investigation, J.G.H.; data curation, J.G.H.; writing—original draft preparation, J.G.H.; writing—review and editing, E.M.C.K., R.S.-R., V.M.C.; supervision, E.M.C.K., R.S.-R., V.M.C. All authors have read and agreed to the published version of the manuscript.

**Funding:** This research received no external funding.

**Data Availability Statement:** Data are contained within the article.

**Acknowledgments:** This publication was made possible in part by the support of Vista Energy Company, which provided Vaca Muerta samples, and W. D. Von Gonten Engineering, which provided their laboratory facilities and technical team to conduct all the laboratory tests.

**Conflicts of Interest:** The authors declare no conflict of interest.

## Appendix A. Thermodynamics of Deformable Continua

This section introduces the mathematical theory of plasticity in the context of the thermomechanics of deformable bodies (see [40]). Let $\mathcal{B}$ be a continuum body with volume $V$ and boundary $\partial\mathcal{B}$. Let us define on $\mathcal{B}$ the following scalar fields: $\rho$, $\theta$, $u$, $s$, and $R$, representing the mass density, temperature, internal energy, entropy, and heat density, respectively. We assume that small deformations adequately describe the system's kinematics. Thus, the first (energy conservation) and second (entropy production imbalance in the form of the Clasius–Duhem inequality) thermodynamic laws in their local form adopt the following differential expressions:

1.  Energy conservation:

$$\rho\,\dot{u} = \boldsymbol{\sigma} : \dot{\boldsymbol{\varepsilon}} - \nabla\cdot\boldsymbol{q} + \rho\,R, \tag{A1}$$

where $\boldsymbol{\sigma}, \dot{\boldsymbol{\varepsilon}} \in \mathbb{R}^{3\times 3}$ are Cauchy's stress and strain rate tensors, respectively, $\nabla = \dfrac{\partial}{\partial x_i}\mathbf{e}_i$ is the gradient operator, and $\boldsymbol{q}$ is the heat flux vector acting on $\partial\mathcal{B}$.

2.  Entropy production imbalance:

$$\rho\,\dot{s} + \nabla\cdot\left(\frac{\boldsymbol{q}}{\theta}\right) - \frac{\rho\,R}{\theta} \geq 0. \tag{A2}$$

Substituting (A1) into (A2), we obtain the Clausius–Duhem inequality,

$$\rho\,\dot{s} + \nabla\cdot\left(\frac{\boldsymbol{q}}{\theta}\right) - \frac{1}{\theta}(\rho\,\dot{u} - \boldsymbol{\sigma} : \dot{\boldsymbol{\varepsilon}} + \nabla\cdot\boldsymbol{q}) \geq 0. \tag{A3}$$

We define the Helmholtz free energy per unit of mass as

$$\Psi := u - \theta\,s. \tag{A4}$$

Recalling that (by the distributive property of the $\nabla$ operator over vector and scalar fields),

$$\nabla\cdot\left(\frac{\boldsymbol{q}}{\theta}\right) = \frac{1}{\theta}\nabla\cdot\boldsymbol{q} - \frac{1}{\theta^2}\boldsymbol{q}\cdot\nabla\theta,$$

and the change of variables $\mathbf{g} = \nabla\theta$, we rewrite (A3) as

$$\boldsymbol{\sigma} : \dot{\boldsymbol{\varepsilon}} - \rho\left(\dot{\Psi} + \dot{\theta}\,s\right) - \frac{1}{\theta}\boldsymbol{q}\cdot\mathbf{g} \geq 0. \tag{A5}$$

**Assumption A1** (Thermodynamic evolution). *We assume that the system's evolution is isothermal, leading to a purely mechanical formulation for* (A5),

$$\boldsymbol{\sigma} : \dot{\boldsymbol{\varepsilon}} - \rho\dot{\Psi} \geq 0. \tag{A6}$$

**Remark A1.** *Every mechanical system must satisfy the Clausius–Duhem inequality of* (A6), *where the system's energy in thermodynamic equilibrium is characterized by a series set of state variables.*

**Axiom A1** (Local-state postulate). *Given an arbitrary system evolution, the energetic state of a continuum medium is characterized by the same state variables that are fixed at the equilibrium state and rate-independent. Therefore, the Helmholtz free energy* (A4) *is given by*

$$\Psi = \tilde{\Psi}(\varepsilon, \boldsymbol{\eta}), \tag{A7}$$

*where* $\boldsymbol{\eta} = \eta_k, \forall k \in \mathbb{N}^+$ *is an internal variable set describing the material's dissipation, and its free energy rate is*

$$\dot{\Psi} = \frac{\partial \Psi}{\partial \varepsilon} : \dot{\varepsilon} + \frac{\partial \Psi}{\partial \boldsymbol{\eta}} * \dot{\boldsymbol{\eta}}. \tag{A8}$$

*The symbol* $" * "$ *denotes the contraction compatible with* $\dfrac{\partial \Psi}{\partial \boldsymbol{\eta}}$ *and* $\dot{\boldsymbol{\eta}}$.

## Appendix B. Reversible and Irreversible Thermodynamic Processes: Elastoplasticity

We characterize the deformation evolution of a material as reversible (non-dissipative) and irreversible (dissipative) by formalizing these concepts using the following definitions:

**Definition A1** (Reversibility: elasticity). *A deformation process is non-dissipative, reversible, or elastic if and only if the internal variables remain constant throughout the deformation process (i.e.,* $\dot{\boldsymbol{\eta}} = 0$*). Thus,* (A6) *reduces to*

$$\boldsymbol{\sigma} : \dot{\varepsilon} - \frac{\partial \Psi}{\partial \varepsilon} : \dot{\varepsilon} = 0. \tag{A9}$$

*Letting* $\Psi = \mathcal{W}$, *where* $\mathcal{W}$ *is the strain energy density, we express the hyperelastic constitutive models as*

$$\boldsymbol{\sigma} = \frac{\partial \mathcal{W}}{\partial \varepsilon}. \tag{A10}$$

We define dissipative thermodynamic processes by adopting an additive decomposition of the strain tensor.

**Assumption A2** (Strain-tensor additive decomposition). *Let* $\varepsilon \in \mathbb{R}^{3 \times 3}$ *be the strain tensor at a material point* $\boldsymbol{x} \in \mathcal{B}$, *which admits the following decomposition:*

$$\varepsilon = \varepsilon^e + \varepsilon^p, \tag{A11}$$

*where* $\varepsilon^e$ *and* $\varepsilon^p$ *are the elastic and plastic components of the strain tensor, respectively.*

**Definition A2** (Dissipative Processes). *A thermodynamic deformation process is dissipative, irreversible, or plastic if the free energy u can be decomposed in a strain energy density* $\mathcal{W}$ *and a latent energy density* $\mathcal{V}$. *Thus, defining* $\Psi = \tilde{\Psi}(\varepsilon, \varepsilon^p, \boldsymbol{\eta})$ *and considering Assumption* A2,

$$\tilde{\Psi}(\varepsilon, \varepsilon^p, \boldsymbol{\eta}) = \mathcal{W}(\varepsilon - \varepsilon^p) + \mathcal{V}(\boldsymbol{\eta}) = \mathcal{W}(\varepsilon^e) + \mathcal{V}(\boldsymbol{\eta}).$$

*Therefore, the Clausius–Duhem inequality* (A6) *becomes*

$$\boldsymbol{\sigma} : \dot{\varepsilon} - \frac{\partial \mathcal{W}}{\partial \varepsilon^e} : \dot{\varepsilon}^e - \frac{\partial \mathcal{V}}{\partial \boldsymbol{\eta}} * \dot{\boldsymbol{\eta}} \geq 0$$

*where* $\dot{\varepsilon} = \dot{\varepsilon}^e + \dot{\varepsilon}^p$ *and* $\boldsymbol{\sigma} = \dfrac{\partial \mathcal{W}}{\partial \varepsilon^e}$. *Thus,*

$$\boldsymbol{\sigma} : \dot{\varepsilon}^p - \frac{\partial \mathcal{V}}{\partial \boldsymbol{\eta}} * \dot{\boldsymbol{\eta}} \geq 0. \tag{A12}$$

**Remark A2** (Reversible and irreversible processes). *A system's evolution is reversible if and only if there exists an isomorphism between the initial and the final state; otherwise, the evolution of a system is irreversible.*

For irreversible processes, the Clausius–Duhem inequality is satisfied for more than one set of state variables. Thus, we enforce uniqueness using the maximum plastic dissipation condition.

**Proposition A1** (Maximum Plastic-Dissipation Condition). *Let $\mathcal{D}^p : \mathbb{R}^{3\times3} \times \mathbb{R}^n \times \mathbb{R}^{3\times3} \mapsto \mathbb{R}$ be the plastic dissipation:*

$$\mathcal{D}^p(\boldsymbol{\sigma}, \dot{\boldsymbol{\eta}}, \dot{\boldsymbol{\varepsilon}}^p) := \boldsymbol{\sigma} : \dot{\boldsymbol{\varepsilon}}^p - \frac{\partial \mathcal{V}}{\partial \boldsymbol{\eta}} * \dot{\boldsymbol{\eta}}. \tag{A13}$$

*Let $\mathbb{E}_\sigma$ be a closed and convex set defining the admissible state variables $\mathbb{E}_\sigma$ given by*

$$\mathbb{E}_\sigma := \left\{ (\boldsymbol{\tau}, \dot{\boldsymbol{\kappa}}) \in \mathbb{R}^{3\times3} \times \mathbb{R}^n, F_f(\boldsymbol{\tau}, \dot{\boldsymbol{\kappa}}) \leq 0 \right\}, \tag{A14}$$

*where $F_f : \mathbb{R}^{3\times3} \times \mathbb{R}^n \mapsto \mathbb{R}$ bounds the admissible stress states (yield function). This assumption defines a unique state variable set $(\boldsymbol{\sigma}, \dot{\boldsymbol{\eta}}, \dot{\boldsymbol{\varepsilon}}^p)$ such that*

$$\mathcal{D}^p(\boldsymbol{\sigma}, \dot{\boldsymbol{\eta}}, \dot{\boldsymbol{\varepsilon}}^p) = \arg\max_{(\boldsymbol{\tau}, \dot{\boldsymbol{\kappa}}) \in \mathbb{E}_\sigma} \mathcal{D}^p(\boldsymbol{\tau}, \dot{\boldsymbol{\kappa}}, \dot{\boldsymbol{\varepsilon}}^p). \tag{A15}$$

**Remark A3** (Elastic Domain and Flow Surface). *The set of admissible states $\mathbb{E}_\sigma$ admits a partition $\mathbb{E}_\sigma = \mathrm{int}(\mathbb{E}_\sigma) \bigcup \partial\mathbb{E}_\sigma$, where $\mathrm{int}(\mathbb{E}_\sigma)$ is the elastic domain defined by*

$$\mathrm{int}(\mathbb{E}_\sigma) := \left\{ (\boldsymbol{\sigma}, \dot{\boldsymbol{\eta}}) \in \mathbb{R}^{3\times3} \times \mathbb{R}^n, F_f(\boldsymbol{\sigma}, \dot{\boldsymbol{\eta}}) < 0 \right\},$$

*and $\partial\mathbb{E}_\sigma$ is the flow surface defined as*

$$\partial\mathbb{E}_\sigma := \left\{ (\boldsymbol{\sigma}, \dot{\boldsymbol{\eta}}) \in \mathbb{R}^{3\times3} \times \mathbb{R}^n, F_f(\boldsymbol{\sigma}, \dot{\boldsymbol{\eta}}) = 0 \right\}.$$

Proposition A1 finds the state variables that maximize $\mathcal{D}^p$ subject to the constraint $F_f(\boldsymbol{\sigma}) = 0$ (consistency condition) and the complementary Kuhn–Tucker conditions [41]. We solve this optimization problem by introducing a Lagrangian (cost function) that transforms the maximization into a minimization problem (i.e., let $-\mathcal{D}^p(\boldsymbol{\tau}, \dot{\boldsymbol{\kappa}}, \dot{\boldsymbol{\varepsilon}}^p)$)

$$\mathcal{L}(\boldsymbol{\tau}, \dot{\boldsymbol{\kappa}}, \lambda) := \gamma \, F_f(\boldsymbol{\tau}, \dot{\boldsymbol{\kappa}}) - \mathcal{D}^p(\boldsymbol{\tau}, \dot{\boldsymbol{\kappa}}, \dot{\boldsymbol{\varepsilon}}^p),$$

where $\mathcal{L} : \mathbb{R}^{3\times3} \times \mathbb{R}^n \times \mathbb{R}^+ \mapsto \mathbb{R}$ is the Lagrangian and $\gamma \in \mathbb{R}^+$ is the Lagrange multiplier. Henceforth, satisfying the maximum plastic dissipation condition is equivalent to solving the following minimization problem:

$$\begin{cases} \textit{Find } \boldsymbol{\sigma}, \dot{\boldsymbol{\eta}} \in \mathbb{E}_\sigma \textit{ and } \gamma \in \mathbb{R}^+, \textit{ such that} \\ (\boldsymbol{\sigma}, \dot{\boldsymbol{\eta}}, \gamma) := \arg\min_{(\boldsymbol{\tau}, \dot{\boldsymbol{\kappa}}) \in \mathbb{E}_\sigma} \gamma \, F_f(\boldsymbol{\tau}, \dot{\boldsymbol{\kappa}}) - \mathcal{D}^p(\boldsymbol{\tau}, \dot{\boldsymbol{\kappa}}, \dot{\boldsymbol{\varepsilon}}^p). \end{cases} \tag{A16}$$

The necessary optimal condition is $\nabla\mathcal{L}|_{(\boldsymbol{\sigma}, \dot{\boldsymbol{\eta}})} = 0$, and the Lagrangian is convex (concave); thus, we can deduce

$$\frac{\partial \mathcal{L}}{\partial \boldsymbol{\sigma}} = -\dot{\boldsymbol{\varepsilon}}^p + \gamma \frac{\partial F_f}{\partial \boldsymbol{\sigma}} = 0, \tag{A17}$$

$$\frac{\partial \mathcal{L}}{\partial \boldsymbol{\eta}} = \mathcal{H} * \dot{\boldsymbol{\eta}} + \gamma \frac{\partial F_f}{\partial \boldsymbol{\eta}} = 0, \tag{A18}$$

$$\frac{\partial \mathcal{L}}{\partial \gamma} = F_f(\boldsymbol{\sigma}, \dot{\boldsymbol{\eta}}) = 0. \tag{A19}$$

where $\mathcal{H}$ is the hardening modulus given by

$$\mathcal{H} := \frac{\partial^2 \mathcal{V}}{\partial \boldsymbol{\eta} * \partial \boldsymbol{\eta}}.$$

From (A17) and (A18) and considering the following change of variable $\gamma = \dot{\lambda}$, the generalized associative flow rule and the generalized hardening law are as follows[36]:

- Generalized Associative Flow Rule:

$$\dot{\boldsymbol{\varepsilon}}^{\mathrm{p}} = \dot{\lambda} \frac{\partial F_f}{\partial \boldsymbol{\sigma}}. \tag{A20}$$

- Generalized Hardening Law:

$$\dot{\boldsymbol{\eta}} = -\dot{\lambda} \, \mathcal{H}^{-1} \frac{\partial F_f}{\partial \boldsymbol{\eta}}. \tag{A21}$$

**Remark A4** (Non-Associative Flow Rules). *Proposition A1 is a sufficient condition that is too restrictive in general, but it inherently induces a generalized hardening law and a generalized associative flow rule. The associative plastic flow rule appropriately characterizes materials with crystalline micro-structural composition. Although associative flow rules characterize the plastic flow of granular materials, non-associative flow rules could be more suitable. Therefore, the Clausius–Duhem Inequality in its local form should be verified independently when using non-associated plastic flow rules. Typically, a non-associated flow rule is defined as*

$$\dot{\boldsymbol{\varepsilon}}^p := \dot{\lambda} \, \mathcal{G}(\boldsymbol{\sigma}, \boldsymbol{\eta}) \quad \text{where} \quad \mathcal{G}(\boldsymbol{\sigma}, \boldsymbol{\eta}) \neq \frac{\partial F_f}{\partial \boldsymbol{\sigma}}. \tag{A22}$$

*These heuristic definitions seek to reconcile experimental observations with simulations by relaxing the overly restrictive Proposition A1, while their major weakness is their ad hoc nature.*

**Appendix C. Complementary Calculations**

*Appendix C.1. Derivatives of the Modified Cam-Clay Yield Function*

We use a modified Cam-Clay yield function with the following form:

$$F_f(\boldsymbol{\sigma}) = \frac{q^2}{M^2} + p(p - p_c),$$

where $q = \sqrt{\frac{3}{2}} \, \|\mathbf{s}\|$ measures the deviatoric effective stress, $p$ is the hydrostatic stress, and $M$ and $p_c$ are material parameters. We calculate the derivative of $F_f$ with respect to $\boldsymbol{\sigma}$, which, by the chain rule, results in

$$\frac{\partial F_f(\boldsymbol{\sigma})}{\partial \boldsymbol{\sigma}} = \frac{\partial F_f(\boldsymbol{\sigma})}{\partial p} \frac{\partial p}{\partial \boldsymbol{\sigma}} + \frac{\partial F_f(\boldsymbol{\sigma})}{\partial q} \frac{\partial q}{\partial \boldsymbol{\sigma}}. \tag{A23}$$

We obtain the derivatives of $F_f$ with respect to $p$, $q$, and $p_c$ directly if we reformulate $F_f$ as

$$F_f(\boldsymbol{\sigma}) = \frac{q^2}{M^2} + p^2 - p \, p_c.$$

Thus,

$$\frac{\partial F_f(\boldsymbol{\sigma})}{\partial p} = 2p - p_c, \tag{A24}$$

$$\frac{\partial F_f(\boldsymbol{\sigma})}{\partial q} = \frac{2q}{M^2}, \tag{A25}$$

$$\frac{\partial F_f(\boldsymbol{\sigma})}{\partial p_c} = -p. \tag{A26}$$

The derivative of $p$ with respect to the effective stress tensor is collected in the following complementary results.

**Lemma A1.** *Let* $\boldsymbol{\sigma} \in \mathbb{R}^{3\times3}$ *be a second-order tensor and* $\sigma_{ij}$, $i$, $j = 1, 2, 3$ *be its components. Then, the following result holds:*

$$\frac{\partial \sigma_{ij}}{\partial \sigma_{kl}} = \delta_{ik}\delta_{jl}, \quad i, j, k, l = 1, 2, 3.$$

**Proof.** The result follows from the index inspection

$$\frac{\partial \sigma_{ij}}{\partial \sigma_{kl}} = \begin{cases} 1, & \text{for } i = k \text{ and } j = l. \\ 0, & \text{otherwise.} \end{cases}$$

Additionally,

$$\delta_{ik}\delta_{jl} = \begin{cases} 1, & \text{for } i = k \text{ and } j = l. \\ 0, & \text{otherwise.} \end{cases}$$

□

**Proposition A2.** *Let* $p$ *be the trace of Cauchy's effective stress tensor; thus, the following holds:*

$$\frac{\partial p}{\partial \boldsymbol{\sigma}} = \frac{1}{3}\delta_{ij}\,\boldsymbol{e}_i \otimes \boldsymbol{e}_j = \frac{1}{3}\mathbf{1}. \tag{A27}$$

**Proof.** Since $p = \dfrac{1}{3}\boldsymbol{\sigma} : \mathbf{1}$ and using index notation, the derivative with respect to the effective stress tensor is

$$\begin{aligned}
\frac{\partial p}{\partial \boldsymbol{\sigma}} &= \frac{1}{3}\frac{\partial}{\partial \boldsymbol{\sigma}}(\boldsymbol{\sigma} : \mathbf{1}) \\
&= \frac{1}{3}\frac{\partial}{\partial \sigma_{kl}}\left(\sigma_{ij}\,\delta_{ij}\right)\mathbf{e}_k \otimes \mathbf{e}_l \\
&= \frac{1}{3}\frac{\partial \sigma_{ij}}{\partial \sigma_{kl}}\,\delta_{ij}\,\mathbf{e}_k \otimes \mathbf{e}_l \\
&= \frac{1}{3}\delta_{ik}\,\delta_{jl}\,\delta_{ij}\mathbf{e}_k \otimes \mathbf{e}_l \quad \text{(by Lemma A1)} \\
&= \frac{1}{3}\delta_{ij}\,\mathbf{e}_i \otimes \mathbf{e}_j.
\end{aligned}$$

□

The derivative of $q$ with respect to the effective stress tensor is collected in the following results.

**Lemma A2.** *Let* $\sigma$, $s \in \mathbb{R}^{3 \times 3}$ *be a second-order tensor and its deviatoric part. Let* $\sigma_{ij}$ *and* $s_{ij}$, *i, j = 1, 2, 3 be the components of* $\sigma$ *and* $s$, *respectively. The following holds:*

$$\frac{\partial s_{ij}}{\partial \sigma_{kl}} = \delta_{ik}\,\delta_{lj} - \frac{1}{3}\delta_{ij}\,\delta_{kl}.$$

**Proof.** Recall that $s_{ij} = \sigma_{ij} - \frac{1}{3}p\delta_{ij}$; thus,

$$\frac{\partial\,s_{ij}}{\partial\sigma_{kl}} = \frac{\partial}{\partial\sigma_{kl}}\left(\sigma_{ij} - \frac{1}{3}\,p\delta_{ij}\right).$$

By Lemma A1 and Proposition A2 and expanding the left-hand side of the previous equality,

$$\frac{\partial\,s_{ij}}{\partial\,\sigma_{kl}} = \frac{\partial\sigma_{ij}}{\partial\sigma_{kl}} - \frac{1}{3}\frac{\partial\,p}{\partial\sigma_{ij}}\,\delta_{kl}$$

$$= \delta_{ik}\delta_{lj} - \frac{1}{3}\,\delta_{ij}\delta_{kl}.$$

□

**Proposition A3.** *Let* $q = \sqrt{\dfrac{3}{2}}\,\|s\| = \sqrt{\dfrac{3}{2}\,s:s}$ *be a measure for the deviatoric stress tensor; thus, the following holds:*

$$\frac{\partial q}{\partial \sigma} = \sqrt{\frac{3}{2}}\,\frac{s}{\|s\|}.$$

**Proof.** We expand the left-hand side of the equality using index notation, the definition of the norm, and double contraction for second-order tensors, as follows:

$$\frac{\partial q}{\partial\sigma} = \frac{\partial q}{\partial\sigma_{ij}}\,\mathbf{e}_i \otimes \mathbf{e}_j$$

$$= \underbrace{\frac{\partial}{\partial\sigma_{ij}}\left(\sqrt{\frac{3}{2}s_{kl}\,s_{kl}}\right)}_{(A)}\,\mathbf{e}_i \otimes \mathbf{e}_j.$$

Applying the chain rule for derivatives in (A), we obtain

$$\frac{\partial}{\partial\sigma_{ij}}\left(\sqrt{\frac{3}{2}s_{kl}\,s_{kl}}\right) = \sqrt{\frac{3}{2}}\,\frac{\partial}{\partial\sigma_{ij}}\,(s_{kl}\,s_{kl})^{\frac{1}{2}}$$

$$= \sqrt{\frac{3}{2}}\left[\frac{1}{2}\left(s_{pq}\,s_{qp}\right)^{-\frac{1}{2}}\frac{\partial}{\partial\,\sigma_{ij}}\,(s_{kl}\,s_{kl})\right]$$

$$= \sqrt{\frac{3}{2}}\left\{\frac{1}{2}\left(s_{pq}\,s_{pq}\right)^{-\frac{1}{2}}\left[\frac{\partial\,s_{kl}}{\partial\sigma_{ij}}\,s_{kl} + s_{kl}\,\frac{\partial\,s_{kl}}{\partial s_{kl}}\right]\right\}$$

$$= \sqrt{\frac{3}{2}}\,(s_{pq}\,s_{pq})^{-\frac{1}{2}}\left(\delta_{ik}\,\delta_{lj} - \frac{1}{3}\,\delta_{ij}\,\delta_{kl}\right)s_{kl} \qquad \text{(by Lemma A2)}$$

$$= \sqrt{\frac{3}{2}}\,(s_{pq}\,s_{pq})^{-\frac{1}{2}}\left(\sigma_{ij} - \frac{1}{3}\sigma_{ll}\,\delta_{ij}\right)$$

$$= \sqrt{\frac{3}{2}}\,(s_{pq}\,s_{pq})^{-\frac{1}{2}}\left(\sigma_{ij} - p\,\delta_{ij}\right) \qquad \left(\text{since } p = \frac{1}{3}\sigma_{ll} = \frac{1}{3}\text{tr}(\sigma)\right).$$

Replacing (A) and contracting the indexes, we obtain the desired expression.

$$\frac{\partial q}{\partial \boldsymbol{\sigma}} = \sqrt{\frac{3}{2}} \left(s_{pq}\, s_{pq}\right)^{-\frac{1}{2}} \left(\sigma_{ij} - p\, \delta_{ij}\right) \mathbf{e}_i \otimes \mathbf{e}_j$$
$$= \sqrt{\frac{3}{2}} \frac{s_{ij}}{\|\mathbf{s}\|} \mathbf{e}_i \otimes \mathbf{e}_j$$
$$= \sqrt{\frac{3}{2}} \frac{\mathbf{s}}{\|\mathbf{s}\|}.$$

$\square$

Therefore, the derivatives of the modified Cam-Clay yield function are collected in the following.

**Proposition A4** (Cam-Clay Yield Function Derivative). *Let $F_f(\boldsymbol{\sigma}) : \mathbb{R}^{3 \times 3} \mapsto \mathbb{R}$ be the Cam-Clay yield function. Then,*

$$F_f(\boldsymbol{\sigma}) := \frac{q^2}{M^2} + p\,(p - p_c), \ \forall M,\ p_c > 0.$$

*Thus, the derivative of $F_f$ with respect to the effective stress tensor adopts the following expression:*

$$\frac{\partial F_f(\boldsymbol{\sigma})}{\partial \boldsymbol{\sigma}} = \frac{1}{3}\,(2p - p_c)\,\mathbf{1} + \sqrt{\frac{3}{2}}\,\frac{2q}{M^2}\,\frac{\mathbf{s}}{\|\mathbf{s}\|}.$$

**Proof.** The result follows by applying the chain rule for derivation,

$$\frac{\partial F_f(\boldsymbol{\sigma})}{\partial \boldsymbol{\sigma}} = \frac{\partial F_f(\boldsymbol{\sigma})}{\partial p}\,\frac{\partial p}{\partial \boldsymbol{\sigma}} + \frac{\partial F_f(\boldsymbol{\sigma})}{\partial q}\,\frac{\partial q}{\partial \boldsymbol{\sigma}},$$

and replacing the corresponding terms with (A24), (A25), and the results from Propositions A2 and A3. $\square$

*Appendix C.2. Discrete Deviatoric Measure*

Let $\boldsymbol{\sigma}_{n+1}^k$ be the updated effective stress update of (36). The updated deviatoric effective stress tensor is

$$\mathbf{s}_{n+1}^k = \boldsymbol{\sigma}_{n+1}^k - p_{n+1}^k\,\mathbf{1}. \tag{A28}$$

Replacing (36) and (38) in (A28), the following holds:

$$\mathbf{s}_{n+1}^k = \hat{\boldsymbol{\sigma}}_{n+1} - \mathbb{C}^e : \Delta\boldsymbol{\varepsilon}^p - \hat{p}_{n+1}\mathbf{1} + K\,\Delta\varepsilon_v^p\,\mathbf{1}$$
$$= \hat{\mathbf{s}}_{n+1} - \underbrace{\mathbb{C}^e : \Delta\boldsymbol{\varepsilon}^p}_{(A)} + K\Delta\varepsilon_v^p\,\mathbf{1}. \tag{A29}$$

$(A)$ can be written as

$$\mathbb{C}^e : \Delta\boldsymbol{\varepsilon}^p = \left[ K\mathbf{1} \otimes \mathbf{1} + 2G\left(\mathbb{I} - \frac{1}{3}\mathbf{1} \otimes \mathbf{1}\right) \right] : \Delta\boldsymbol{\varepsilon}^p$$
$$= K\underbrace{\mathbf{1} \otimes \mathbf{1} : \Delta\boldsymbol{\varepsilon}^p}_{(B)} + \underbrace{2G\,\mathbb{I} : \Delta\boldsymbol{\varepsilon}^p}_{(C)} - \frac{2}{3}G\mathbf{1} \otimes \mathbf{1} : \Delta\boldsymbol{\varepsilon}^p.$$

Now, expanding $(B)$,

$$\Delta \varepsilon^{\mathrm{P}} : \mathbf{1} \otimes \mathbf{1} = \Delta \varepsilon_{ij}^{\mathrm{P}} \, \mathbf{e}_i \otimes \mathbf{e}_j : \delta_{kl} \, \mathbf{e}_k \otimes \mathbf{e}_l \otimes \delta_{pq} \, \mathbf{e}_p \otimes \mathbf{e}_q$$
$$= \Delta \varepsilon_{ij}^{\mathrm{P}} \, \delta_{ik} \delta_{jl} \, \delta_{pq} \mathbf{e}_p \otimes \mathbf{e}_q$$
$$= \Delta \varepsilon_{kk}^{\mathrm{P}} \, \delta_{pq} \mathbf{e}_p \otimes \mathbf{e}_q$$
$$= \Delta \varepsilon_{\mathrm{v}}^{\mathrm{P}} \, \mathbf{1}.$$

Considering the definition of $\mathbb{I}$ in (6), $(C)$ admits the following expansion:

$$2G \, \mathbb{I} : \Delta \varepsilon^{\mathrm{P}} = 2G \left[ \frac{1}{2} \left( \delta_{ik} \, \delta_{jl} + \delta_{il} \, \delta_{jk} \right) \Delta \varepsilon_{ij}^{\mathrm{P}} \right]$$
$$= 2G \left[ \frac{1}{2} \left( \Delta \varepsilon_{kl}^{\mathrm{P}} + \Delta \varepsilon_{lk}^{\mathrm{P}} \right) \right]$$
$$= 2G \, \Delta \varepsilon^{\mathrm{P}} \qquad \text{(thanks to the symmetry of } \Delta \varepsilon^{\mathrm{P}} \text{).}$$

Replacing $(B)$ and $(C)$ in $(A)$ and then replacing in (A29), we obtain the following expression:

$$\mathbf{s}_{n+1}^k = \hat{\mathbf{s}}_{n+1} - \cancel{K \Delta \varepsilon_{\mathrm{v}} \mathbf{1}} - 2G \, \Delta \varepsilon^{\mathrm{P}} + \frac{2}{3} G \Delta \varepsilon_{\mathrm{v}}^{\mathrm{P}} \mathbf{1} + \cancel{K \Delta \varepsilon_{\mathrm{v}}^{\mathrm{P}} \mathbf{1}}$$
$$= \hat{\mathbf{s}}_{n+1} - 2G \left( \Delta \varepsilon^{\mathrm{P}} - \frac{1}{3} \Delta \varepsilon_{\mathrm{v}}^{\mathrm{P}} \mathbf{1} \right)$$
$$= \hat{\mathbf{s}}_{n+1} - 2G \, \Delta \varepsilon_{\mathrm{d}}^{\mathrm{P}}, \tag{A30}$$

where $\Delta \varepsilon_{\mathrm{d}}^{\mathrm{P}} = \Delta \varepsilon^{\mathrm{P}} - \frac{1}{3} \Delta \varepsilon_{\mathrm{v}}^{\mathrm{P}} \, \mathbf{1}$. By the definition of the updated deviatoric measure $q_{n+1}^k$, the following holds:

$$q_{n+1}^k = \sqrt{\frac{3}{2}} \| \hat{\mathbf{s}}_{n+1} - 2G \, \Delta \varepsilon_{\mathrm{d}}^{\mathrm{P}} \|. \tag{A31}$$

From the discrete flow rule (37) and the derivative of $F_f(\boldsymbol{\sigma})$ with respect to $\boldsymbol{\sigma}$, we deduce that $\hat{\mathbf{s}}_{n+1}$, $\mathbf{s}_{n+1}^k$, and $\Delta \varepsilon_{\mathrm{d}}^{\mathrm{P}}$ are co-linear. Thus,

$$q_{n+1}^k = \sqrt{\frac{3}{2}} \| \hat{s}_{n+1} \| - 2G \sqrt{\frac{3}{2}} \| \Delta \varepsilon^{\mathrm{P}} \|$$
$$= \hat{q}_{n+1} - 3G \Delta \varepsilon^{\mathrm{P}}, \tag{A32}$$

where $\| \Delta \varepsilon^{\mathrm{P}} \| = \Delta \lambda \sqrt{\frac{3}{2}} \dfrac{2 q_{n+1}^k}{M^2}$.

*Appendix C.3. Derivative of Modified Cam-Clay Yield Function and Hardening Function with Respect to $\Delta \lambda$*

We fully define the iterative scheme for determining the discrete consistency parameter $\Delta \lambda$ using the following.

**Proposition A5** (Derivative of $F_f$ with respect to $\Delta \lambda$). *Let $F_f(\Delta \lambda)$ be given by (41) and consider (39), (41), (42), (A24), (A25) and (A26). Then, the following result holds:*

$$\frac{\partial F_f}{\partial \Delta \lambda} = -K \frac{(2 \, p - p_c)^2}{1 + (2K + \chi \, p_c) \Delta \lambda} - \frac{2q}{M^2} \frac{q}{\Delta \lambda + \frac{M^2}{6G}} - \chi \, p \, p_c \frac{(2p - p_c)}{1 + (2K + \chi \, p_c) \Delta \lambda}. \tag{A33}$$

**Proof.** By the chain rule, we obtain

$$\frac{\partial F_f}{\partial \Delta\lambda} = \frac{\partial F_f}{\partial p}\frac{\partial p}{\partial \Delta\lambda} + \frac{\partial F}{\partial q}\frac{\partial q}{\partial \Delta\lambda} + \frac{\partial p_c}{\partial \Delta\lambda}. \tag{A34}$$

The derivatives of $F_f$ with respect to $p$, $q$, and $p_c$ are given by (A24), (A25), and (A26). Thus, we only need to calculate $\frac{\partial p}{\partial \Delta\lambda}$, $\frac{\partial q}{\partial \Delta\lambda}$, and $\frac{\partial p_c}{\partial \Delta\lambda}$. Considering (39), (41), and (42), we have

$$\frac{\partial p}{\partial \Delta\lambda} = -K\frac{(2p - p_c)}{1 + (2K + \chi\,p_c)\Delta\lambda}, \tag{A35}$$

$$\frac{\partial q}{\partial \Delta\lambda} = -\frac{q}{\Delta\lambda + \frac{M^2}{6G}}, \tag{A36}$$

$$\frac{\partial p_c}{\partial \Delta\lambda} = \chi p_c\frac{(2p - p_c)}{1 + (2K + \chi p_c)\Delta\lambda}. \tag{A37}$$

The result follows after substituting (A24)–(A26), (A35), (A36), and (A37) into (A34). □

**Proposition A6** (Derivative of $H_f$ with respect to $\Delta\lambda$)**.** *Let $H_f(\Delta\lambda)$ be given by* (43)*. Then, the following result holds:*

$$\frac{\partial H_f}{\partial \Delta\lambda} = -\frac{(p_c)_n \chi \Delta\lambda}{1 + 2\Delta\lambda\,K}\exp\left[\frac{\chi\,(2\,\hat{p}_{n+1} - p_c)}{1 + 2\Delta\lambda\,K}\right] - 1. \tag{A38}$$

**Proof.** Obtain (43) and perform the chain rule for derivation with respect to $\Delta\lambda$. □

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
