# Peer review of "Shear-Enhanced Compaction Analysis of the Vaca Muerta Formation"

_computation, doi:10.3390/computation11120250_

Round 1

Reviewer 1 Report

Comments and Suggestions for Authors

The goal of this paper is

1) to present the triaxial tests on Vaca Muerta formation and

2) calibrate a simple model which matches the observations.

Triaxial test results are well presented and may be useful for model calibration. Hydrostatic compaction test would be a valuable addition to the data set. Yet, some interpretations of these data are not convincing. For example,

in the pictures presented (Fig11) shear-enhanced compaction is associated to the deviation from the linear extrapolation (hydrostatic line). It is not clear if the hydrostatic compaction test would follow this linear line and most likely would bend similar to the other triaxial tests but to a lesser degree.

Also, this interpretation contradicts the assumption of incomprehensibility of the solid material (Assumption #3) in case V_s=const the pressure would not change after the onset of compaction (P >P_c) and may even decrease since the porosity is decreasing and the average pressure can be presented as P=(1-fi)P_s where P_s=P_s(V_s)=const since V_s=const. The plots (c) in Fig.8-11

show otherwise which is the proof that V_s is decreasing as well as V_v.

Assumption by Eq.(36) : K=p/(1-fi) does not have much physical sense since K=dP/d Ev so dP/dEv=aP/(1-fi), after integration gives an exponential growth of pressure at high compression and zero bulk modulus at the initial pressure (p=0) . It makes more sense to present K=k_0+k_1*e_v +.. as function of volumetric compression. Also, according to Eq.(36) bulk modulus increases with increased porosity which is opposite to what is typically observed.

The main novelty in MCC model calibrated , as it follows from the statements in introduction and conclusion (lines 362), is the porous compaction model and nonlinear elasticity (lines 27-29).

But in my opinion both these points are not well described and defended in this paper.

Theoretical background is disproportionately long. The closest Point projection mapping is well described. The limitations of the model are not mentioned. In my view, they include

the fact that the model is purely mechanical without temperature or energy dependence, also effects of fluid saturation of porosity is not discussed. Yet, the presence of fluid in pores will make the Assumption#3 is even less relevant.

Conclusion is disproportionately small, so I suggest to add all this there.

In short I suggest the following :

1) Please, try to get rid of Assumption#3 and Eq.(36). Try to use a reasonable compaction law, for example Fi=Fi_0*(exp S(Pc-P)/Y) (which follows from the pressure equilibrium around an isolated pore in the plastic medium of strength Y) .

2) Experimental part can be supplemented by a hydrostatic compaction test if possible.

3) Add limitations of the model into conclusions and future work

Author Response

Dear reviewer,

Reviewer 2 Report

Comments and Suggestions for Authors

After a careful reading of the manuscript, I can state the following:

It is a potentially interesting paper for readers, but it has not yet reached the quality for publication. A major correction of the manuscript is necessary.

First of all, please explain why you consider it sufficient to carry out a triaxial test with the same confining pressure on only four rock samples and to model the behaviour of the entire Vaca Muerta Formation on this basis? It can only be a kind of preliminary communication before large-scale tests are carried out.

SI units of measurement must be used in the manuscript!

Although the manuscript is extensive, it is partly unreviewed and has no basic discussion chapter! It is imperative that this is corrected. I instruct the authors to redesign the manuscript so that the chapters are as similar as possible to those in the template for this journal.

A discussion chapter must be added. In it, the authors should discuss the results and explain how they can be interpreted in relation to previous studies and the working hypotheses. In this context, you have provided an estimate of Young's Modulus and Poisson's ratio in Table 3. Poisson's ratio is generally in a small range for rocks, so no further explanation is needed, but you have not sufficiently considered other ways of estimating the elastic modulus. Since these properties are very important for further modelling, you should write a few sentences more about the types of estimation in general in the Discussion chapter. As a suggestion, you can use the paper https://doi.org/10.3390/app11136148, which you can include in the references.

The modelling was done in the porous medium and the possibility of a modern method to determine the porosity was not mentioned at all. It is not enough to cite reference no. 31 from 2004, more modern papers such as https://doi.org/10.3390/su13147668 should also be cited.

I suggest that you search the Computation journal database for the keywords "elastoplastic model" and "numerical plasticity" to find some other papers to compare with your results.

The results and their implications should be discussed in as broad a context as possible. For example, you must point out that these studies and modelling only refer to the part of the rock mass of Vaca Muerta that we call rock material. The discontinuity properties that most determine the behaviour of the rock mass are not considered.

Future research directions can also be pointed out (for this you can use part of the current chapter 7. Conclusions and future work).

Minor errors should also be corrected:

Line 229 - Include in the references the ASTM standards to which the test was conducted. Include a picture of the device used during the test and indicate the type of device!

Table 1. The name of the table should be above the table and not below it as indicated in the template. The units of measurement must be given in the system SI, i.e. mm, mm3 should be used and the notation for grammes is only g, not gr!

Line 241. - add a mark for porosity in the explanation of formula (47).

Table 3. - delete the square bracket to the left of the table! In the table, the values must be given in GPa.

References are not written correctly according to the template, e.g. 31!

Comments on the Quality of English Language

In my opinion, the English language is fine, although some minor mistakes need to be corrected.

Author Response

Dear reviewer,

Reviewer 3 Report

Comments and Suggestions for Authors

In the manuscript, a complete laboratory analysis of samples from the Vaca Muerta formation showing experimental evidence of nonlinear elastic and unrecoverable shear-enhanced compaction was presented. An elastoplastic constitutive model using these experimental observations was calibrated. This manuscript is interesting. However, a major revision should be provided by the authors before further consideration. In particular, I attach the comments that in my opinion should be taken into account to improve the manuscript.

1 The presentation of the manuscript is too detailed, especially section 2 and Section 3 If it were a part of the book, it would be a pleasing thing to read. For the paper, please simplify some of the common knowledge or put it in an appendix.

2 The manuscript has a detailed description of the elastoplastic constitutive model; however, the main improvements of the paper are not clear. Please further clarify the innovations of the manuscript.

3. In the introduction, please add an overview of the application of the MCC model to rocks.

4 A review of research on the nonlinear constitutive model of rocks needs to be added and discussed.

5 The formatting of the manuscript needs to be further improved, e.g., please use a three-line table for the tables.

Comments on the Quality of English Language

In the manuscript, a complete laboratory analysis of samples from the Vaca Muerta formation showing experimental evidence of nonlinear elastic and unrecoverable shear-enhanced compaction was presented. An elastoplastic constitutive model using these experimental observations was calibrated. This manuscript is interesting. However, a major revision should be provided by the authors before further consideration. In particular, I attach the comments that in my opinion should be taken into account to improve the manuscript.

1 The presentation of the manuscript is too detailed, especially section 2 and Section 3 If it were a part of the book, it would be a pleasing thing to read. For the paper, please simplify some of the common knowledge or put it in an appendix.

2 The manuscript has a detailed description of the elastoplastic constitutive model; however, the main improvements of the paper are not clear. Please further clarify the innovations of the manuscript.

3. In the introduction, please add an overview of the application of the MCC model to rocks.

4 A review of research on the nonlinear constitutive model of rocks needs to be added and discussed.

5 The formatting of the manuscript needs to be further improved, e.g., please use a three-line table for the tables.

Author Response

Dear Reviewer,

Round 2

Reviewer 2 Report

Comments and Suggestions for Authors

Dear authors, although your "Response to the Reviewer" is not in the usual form for MDPI, I have nevertheless understood and checked in the manuscript that you have made the necessary changes. Your manuscript is therefore suitable for publication in a journal. I wish you every success in your professional and scientific work.